# A Behavior-first approach to Multi-Intention Inverse Reinforcement Learning

## Abstract

Inverse Reinforcement Learning (IRL) seeks to infer reward functions from expert demonstrations. When demonstrations come from multiple experts with different intentions, the problem becomes Multi-Intention IRL (MI-IRL). MI-IRL methods generally follow a reward-first paradigm: they ask which reward, if followed, could have generated each trajectory. This leads to trajectory similarity being based on reward likelihood rather than on the relationships between behaviors. In deep generative MI-IRL methods, this paradigm further couples behavior clustering and reward learning, making such methods dependent on prior knowledge of the true number of modes $K^*$. This limits adaptability to unseen behaviors, and restricts the analysis to learned rewards rather than relationships across behaviors. In contrast, we approach MI-IRL from a different standpoint. We propose Contrastive Multi-Intention IRL (CoMI-IRL), a transformer-based unsupervised behavior-first framework: it first learns a representation of trajectory dynamics that captures intrinsic behavioral similarity, then clusters trajectories in that space, and finally learns a reward for each cluster. Our experiments show that CoMI-IRL outperforms existing approaches without needing $K^*$ or labels, while allowing for visual interpretation of behavior relationships and adaptation to unseen behavior without full retraining.

## 1 Introduction

Inverse Reinforcement Learning (IRL) deals with tasks where the reward function is unknown or difficult to define by recovering a reward function from demonstrations (Ng & Russell, 2000). IRL has progressed from modeling rewards as linear combination of features (Ng & Russell, 2000; Abbeel & Ng, 2004), through ambiguity reduction methods (Ziebart et al., 2008; Ramachandran & Amir, 2007), to deep learning-based methods in generative adversarial (Ho & Ermon, 2016; Fu et al., 2017) settings. However, these approaches assume all demonstrations share a unique goal. This is often not the case, as datasets can contain non-optimal and conflicting modes of behavior collected from multiple sources. Such problem is known as Multi-Intention IRL (MI-IRL) (Babes et al., 2011).

MI-IRL addresses reward modeling in three ways: (i) as a linear combination of features in Expectation-Maximization (EM) (Babes et al., 2011; Bighashdel et al., 2023), (ii) through Bayesian methods (Choi & Kim, 2012; Michini & How, 2012), and (iii) as neural networks in generative frameworks (Li et al., 2017; Fu et al., 2023; Hausman et al., 2017), with the latter achieving most promising results. All three ways ask which reward function, if followed, would most likely have generated each trajectory. In this *reward-first* paradigm, trajectory similarity is defined through reward likelihood rather than through intrinsic relationships between behaviors. In deep generative methods, this paradigm leads to the coupling of behavior clustering and reward learning: a latent code $c$, sampled from a categorical distribution with $K$ fixed categories, simultaneously defines behavioral modes and conditions reward learning, such that the two tasks cannot be disentangled.

This coupling, inherent to reward-first approaches, introduces three key limitations. First, fixing $K$ assumes knowledge of the true number of behavioral modes $K^*$, often unavailable and hard to specify, akin to the challenge of selecting cluster numbers in data analysis (Wang, 2010; Tibshirani et al., 2001; Pan et al., 2013), which can lead to the risk of misspecification. In continual settings, where new behaviors are added, models

with fixed $K$ require full retraining. Methods inferring $K$ from data have been explored (e.g.,nonparametric Bayesian IRL (Choi & Kim, 2012)), however their application in continuous spaces is challenging due to computational limitations (Ramponi et al., 2020). Second, in coupled approaches the failure in one task can cascade and complicate diagnosis (i.e. it's hard to verify if the model failed to learn the reward or to cluster behavior correctly). Finally, this design hinders interpretability, since coupled methods allow to analyze behaviors only by deploying the latent-conditioned policies, unrelated to the original dynamics of the dataset. As these limitations are a byproduct of the reward-first paradigm, we explore an alternative solution.

In this paper, we adopt a *behavior-first* paradigm on MI-IRL, asking which trajectories share similar dynamics rather than which reward function would most likely have generated them. This leads to a decoupled perspective, separating "what happened" (behavior clustering), from "why it happened" (reward learning). We propose Contrastive Multi-Intention IRL (CoMI-IRL), a transformer-based[1] unsupervised framework that learns trajectory embeddings via contrastive training. The learned representations are grouped for their intrinsic patterns, not by reward likelihood. By inducing a similarity structure in the embedding space, we can discover clusters of behavioral modes via graph-based connectivity without knowing $K$. This reduces MI-IRL to independent single-intention IRL per cluster, also enabling direct adaptation to new behaviors.

In our experiments, we analyze how the relationship between the assumed ($K$) and true ($K^*$) number of behavioral modes (i.e., $K \neq K^*, K = K^*$) impacts clustering and reward learning. We show how our approach avoids this problem, facilitates the interpretation of relationships in the dataset and adapts to new behaviors without full retraining. Our results demonstrate that our design outperforms other approaches in both clustering quality and reward learning without requiring $K$, alleviating the limitations of coupled methods, enhancing MI-IRL adaptability to new behaviors.

## 2   Related Work

In this Section, we provide an overview of the current reward-first methods in the field of MI-IRL, and we further explore other methods that also started to look at similar tasks from a behavior-first perspective, especially in the area of trajectory embedding.

**EM-based Methods** Early parametric EM-based methods (Babes et al., 2011; Snoswell et al., 2021; Ramponi et al., 2020) typically require knowledge of $K$ and model the reward function as a linear combination of handcrafted features (Ng & Russell, 2000; Levine et al., 2010), and the performance highly depends on how well these are selected (Choi & Kim, 2013). In CoMI-IRL, as in other deep MI-IRL methods, we do not need to handcraft features.

**Nonparametric Bayesian Methods** When $K$ is unknown, it can be inferred through Bayesian solutions (Choi & Kim, 2012; Michini et al., 2015; Bighashdel et al., 2021), with some works learning the reward more flexibly through Gaussian Process kernels (Levine et al., 2011) or deep learning (Bighashdel et al., 2021). However, this comes at the cost of a significantly larger overhead and computational costs (Ramponi et al., 2020; Wulfmeier et al., 2015; Zhu et al., 2023). The application in continuous state-action spaces as the ones considered in our work is sparsely addressed in literature (Almingol & Montesano, 2015; Rajasekaran et al., 2017) due to the computational limitations of applying Bayesian methods in such spaces. In CoMI-IRL, we work specifically on continuous state-action spaces, where the application of current methods in this line area is limited.

**Deep Generative MI-IRL Methods** Deep Generative MI-IRL method transfer the IRL problem to multi-expert settings by learning multiple rewards, often via latent conditioning. Different methods extends GAIL with MI maximization between trajectories and latent codes $c$, enabling mode separation without explicit clustering (Li et al., 2017; Hausman et al., 2017; Fu et al., 2023). However, maximizing a lower bound on MI in these works depends on the generator's ability to cover the distribution. If the policy fails to capture a specific mode, the clustering signal degrades. In contrast, CoMI-IRL's behavior-first decoupled design employs a discriminative objective that optimizes the embedding geometry for separability, independent of the policy's performance.

---

[1]The repository will be made available on GitHub after acceptance or upon request.

The approaches mentioned so far cluster trajectories via a reward-first likelihood-based mechanisms: in EM-based methods, it is the likelihood of a trajectory being generated by a certain reward function; Bayesian solutions cluster via posterior inference over the components of the mixture model; and generative approaches cluster through the probability of generation from a conditioned policy. A connection between likelihood-based and generative-based approaches in IRL was established by Finn et al. (2016), which persists in the multi-intention case. In contrast, CoMI-IRL clusters via contrastive embedding similarity. In each method, behavioral structure is discovered through reward-generating plausibility rather than through intrinsic dynamic similarity. With CoMI-IRL, instead, we define similarity by trajectory dynamics in a learned embedding space, adopting a behavior-first paradigm.

**Trajectory Embedding Methods** The use of sequence models and contrastive techniques to learn trajectory embeddings gathered a lot of attention in recent works. Vivekanandan et al. (2025) employ Transformer encoders with triplet losses on spatially augmented short trajectories to learn motion priors for forecasting. TrajCL (Chang et al., 2023) employs dual-feature (spatial/structural) self-attention Transformers with contrastive augmentations (shifting/masking) for similarity queries, focusing on spatial ranking without intention handling. Kujanpää et al. (2025) proposes Discrete Style Diffusion Policy (DSDP) as a Behavioral Cloning method, combining InfoNCE with Lookup-Free Quantization (LFQ) to extract discrete styles on sub-trajectories for conditional diffusion behavior cloning, tying embeddings to generation rather than reward inference. Ge et al. (2025) introduce Variational Trajectory Embeddings (VTE), a method designed to capture a continuous ability vector representing a trajectory's proficiency on a single task. In VTE, first a probabilistic skill extractor (based on a Hierarchical State Space Model and the Learning Options via Compression (LOVE) (Jiang et al., 2022) framework) processes a trajectory to yield a sequence of latent skill distributions. Secondly, a transformer-based Variational Autoencoder (VAE) gets this skill sequence in input to compute final embedding's posterior. The model is trained to maximize the VAE's Evidence Lower Bound (ELBO), pairing a KL divergence regularizer and an action reconstruction loss, where a policy conditioned on the embedding must replicate the trajectory's actions.

In contrast with VTE's variational approach, which relies on intermediate skill representations and a reconstruction objective to model how a single task is performed, CoMI-IRL learns directly from raw state-action sequences using a purely contrastive objective. Its goal is to learn a discriminative embedding space for behavioral clustering, enabling a fully decoupled and adaptive MI-IRL method.

## 3 Background

In this section, we formulate the MI-IRL problem and review the key concepts in contrastive learning applied in our work.

### 3.1 Multi-intention IRL

In multi-intention IRL, a finite-state Markov Decision Process without Rewards (MDP\R) is the tuple $(S, A, T, \gamma, b_0, \tau_1, \ldots, \tau_D)$, where $S$ is the state space, $A$ is the action space, $T : S \times A \times S \to [0, 1]$ is the transition probability function, $\gamma \in [0, 1)$ is the discount factor, $b_0$ is the probability of starting in state $s$, and $\tau_i = (s_0, a_0, s_1, a_1, \ldots, s_T, a_T)$ is the $i^{th}$ trajectory in a dataset of $D$ trajectories. Each trajectory is generated from the optimal policy of one of $K^*$ behavioral modes, each corresponding to an intention mathematically represented by a reward function. Trajectories are assumed to be without labels, and the goal is to infer the number of behavioral modes and each respective reward function.

### 3.2 Contrastive Learning

Contrastive Learning (CL) is a self-supervised technique for learning representations by mapping similar (positive) examples closer in an embedding space, and dissimilar (negative) objects apart. CL is often applied to unsupervised sequence embedding methods with pretrained transformers to capture global semantics without requiring labels. For example, BERT (Devlin et al., 2019) prepends a special token called *CLS* to the beginning of every sequence. As the model processes the sequence, this token learns to collect information from all other elements, ending up as a single vector that summarizes the entire input (Qin et al., 2023;

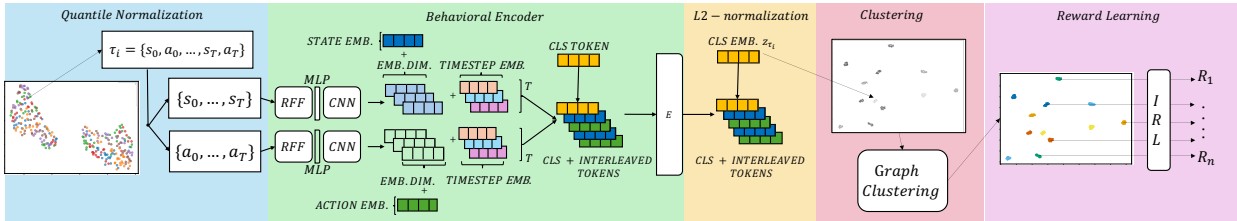

Figure 1: **CoMI-IRL pipeline.** States and actions of each quantile normalized input trajectory $\tau_i$ pass through a Random Fourier Feature (*RFF*) encoding with Gaussian mapping, a shallow *MLP* and a 1D *CNN* layer. After adding timestep and modality embeddings (distinguishing states and actions), we interleave the two sequences and prepend a *CLS* token summarizing the trajectory. On the resulting embeddings, we apply graph-clustering to get cluster labels and apply IRL independently on each cluster.

Wang et al., 2024; Zou et al., 2024). We use this concept to represent trajectories and distinguish behaviors: we treat each $\tau_i$ as a "sentence" and use the *CLS* token as a compact summary of the behavior.

Contrastive learning methods benefits from data augmentation (Deng et al., 2023), but strong augmentations could disrupt behavior coherence. To form positive pairs without disrupting the coherence of the behavior, weak augmentations can be used. An effective strategy to form positive pairs is to create two dropout-augmented views of the input by passing each input sequence through the model twice, while all other trajectories in the batch serve as negative elements, following the intuition from Gao et al. (2021). These dropout-augmented views can then be passed into a contrastive loss such as the symmetric InfoNCE (Oord et al., 2018; Chen et al., 2020b), a central component of our approach that encourages informative and distinguishable embeddings. Since InfoNCE is asymmetric with respect to the anchor, the symmetric version treats each view as an anchor in turn, ensuring both views contribute equally to the contrastive signal. The loss for a single view $z^1_{\tau_i}$ is

$$\ell(z^1_{\tau_i}, z^2_{\tau_i}) = -\log \frac{\exp(\text{sim}(z^1_{\tau_i}, z^2_{\tau_i})/\rho)}{\sum_{k=1, k\neq i}^{2N} \exp(\text{sim}(z^1_{\tau_i}, z^2_{\tau_k})/\rho)}, \tag{1}$$

and symmetric for $z^2_{\tau_i}$, where sim is cosine similarity, $\rho$ is temperature and $N$ is the number of trajectories in the batch. In addition, we want to further distinguish embeddings so that learned representations can be consistently clustered. For this, regularization terms such as a Deep InfoMax (DIM) (Hjelm et al., 2019) can help, by maximizing mutual information via Jensen-Shannon divergence (JSD) so that each local element of the input informs the whole representation. DIM trains a discriminator $f_\phi$ to give positive scores to pairs $f_\phi(z_{\tau_i}(\theta), z_{\tau_i(q)}(\theta))$ with the global summary paired with its own local features (joint pairs), and negative scores to pairs $f_\phi(z_{\tau_i}(\theta), z_{\tau_j(q)}(\theta))$ where $i \neq j$, so that:

$$\mathcal{L}_{\text{DIM}} = -\mathbb{E}_{P_J}[\log \sigma(f_\phi(z_{\tau_i}, z_{\tau_i(q)}))] - \mathbb{E}_{P_M}[\log(1 - \sigma(f_\phi(z_{\tau_i}, z_{\tau_j(q)})))], \tag{2}$$

where $\sigma(x) = \frac{1}{(1+e^{-x})}$ is the sigmoid function.

## 4 Contrastive Multi-Intention IRL

In this section we introduce CoMI-IRL, a behavior-first framework that decouples behavior clustering and reward learning. Using a fully unsupervised contrastive approach, CoMI-IRL maps *quantile normalized* demonstrations into an embedding space via a *Behavioral Encoder (BE)* with *L2-normalized* outputs, followed by *Clustering*, defining single-intention groups for downstream *Reward Learning* (Figure 1). At its core, the BE, a Transformer Encoder-only Vaswani et al. (2017) model with a pre-processing block, generates trajectory representations as *CLS* tokens to represent complex behaviors based on their intrinsic patterns (Section 4.1). As similar trajectories are grouped together, cluster labels are obtained via a graph-based method for the subsequent IRL step (Section 4.2). CoMI-IRL can be adapted to new behavior without full retraining (Section 4.3).

### 4.1 Behavioral Encoder Architecture

The first step is to embed trajectories in a latent space such that similar behaviors are grouped together. We base our encoder-only approach on the Transformer model (Vaswani et al., 2017) as it allows to create representations encoding information between distant timesteps through its attention mechanism. We quantile normalize the input trajectories (Bolstad et al., 2003), independently mapping each dimension to a normal distribution, since the state–action spaces contain high-frequency elements with substantially different magnitudes.

Further, as states $s_{\tau_i}$ and actions $a_{\tau_i}$ of each input trajectory $i-$th have different dynamics, we encode them with two different pre-processing heads, one for each sequence (Chen et al., 2021). Each head applies a Random Fourier Features ($RFF$) encoding with Gaussian mapping (Tancik et al., 2020; Zheng et al., 2022; Li et al., 2021), followed by a shallow $MLP$ and a 1D $CNN$, to capture short-range temporal patterns in the trajectory. The $RFF$ encoding overcomes known $MLP$ spectral bias towards low-frequency functions by injecting a rich set of sinusoidal features at multiple random orientation and frequencies. Each block's output is enriched with a timestep (used as positional encoding) and either a state or action embedding to further differentiate. Next, we interleave them to recreate the single input sequence. We prepend a $CLS$ token to this sequence that will function as the trajectory summary. The resulting sequence is then passed through the encoder to capture the trajectory's global dependencies, summarized in the $CLS$ embedding, which for simplicity we name $z_{\tau_i}$. Finally, we L2-normalize the $CLS$ token of the encoder's output, restricting the output space to the unit hypersphere (Wang & Isola, 2020).

#### 4.1.1 Contrastive Framework and Loss Function

As we aim to learn a discriminative representation $z_{\tau_i}$ of each input trajectory $\tau_i$, we apply the symmetric InfoNCE loss at multiple granularity levels, with DIM included as a regularization term (cf. Section 3.2). First, we generate two dropout-augmented views of each input $\tau_i$: $z_{\tau_i}^1$ and $z_{\tau_i}^2$, forming a positive pair. This maps similar trajectories close together in the embedding space, allowing for the emergence of areas with distinct behavioral modes. This loss component is defined as:

$$\mathcal{L}_{\text{CLS}} = \frac{1}{2N} \sum_{i=1}^{2N} \ell_{\text{InfoNCE}}(z_{\tau_i}^1, z_{\tau_i}^2). \tag{3}$$

Second, we sample $n$ segments of length $L$, with starting point $q \in \{0, T - L\}$ from each input trajectory. Each segment is defined as $\tau_i(q, L) = \{s_q, a_q, ..., s_{q+L}, a_{q+L}\}$, and passed through the encoder to create $n$ segment embeddings $z_{\tau_i(q,L)}^k$. Aiming to promote consistency between local (segment-level) and trajectory representations, we apply InfoNCE between each trajectory embeddings and its segment embeddings. This loss component, $\mathcal{L}_{\text{SEG}}$, performs the same operation as Equation 3, but applied between $z_{\tau_i}$ and $z_{\tau_i(q,L)}^k$. To push representations even closer together in each area, we apply InfoNCE between all unique pairs of segment embeddings:

$$\mathcal{L}_{\text{PAIR}} = \frac{1}{\binom{n}{2}} \sum_{k=1}^{n-1} \sum_{j=k+1}^{n} \ell_{\text{InfoNCE}}(z_{\tau_i(q,L)}^k, z_{\tau_i(q,L)}^j). \tag{4}$$

We also add a DIM component to the loss, $\mathcal{L}_{\text{DIM}}$, which follows Equation 2, with the goal of keeping local structure consistent and further stabilize representations. This component explicitly maximize the mutual information between trajectory representations and their local features without forcing token-level separation across trajectories. Combining the components with $\alpha, \beta, \gamma$ and $\delta$ as balancing coefficients, we get our complete loss:

$$\mathcal{L} = \alpha \mathcal{L}_{\text{CLS}} + \beta \mathcal{L}_{\text{DIM}} + \gamma \mathcal{L}_{\text{SEG}} + \delta \mathcal{L}_{\text{PAIR}}. \tag{5}$$

### 4.2 Clustering and Reward Learning

In principle, any clustering method not requiring $K^*$ could be used. However, density-based methods such as HDBSCAN (Campello et al., 2013) can be highly sensitive to hyperparameters, require clear density gaps

(not often available in continuous manifolds), and its adaptation relies on the fixed density structure of the original tree. Therefore, we cluster the embeddings using a graph-based community detection approach not requiring $K^*$ via local connectivity. We construct a weighted $k$-nearest neighbor graph over trajectory embeddings, where nodes represent trajectories and edge weights are defined by cosine similarity between embedding. This choice is motivated by the fact that our contrastive objective optimizes for directional alignment. Rather than assuming a fixed global scale, we analyze the connectivity structure of the graph to identify stable behavioral partitions. In cases where the graph decomposes into multiple connected components, these components directly define clusters. If the graph remains fully connected , such as in continuous manifolds, we apply community detection using the Leiden algorithm (Traag et al., 2018) to identify coherent subgraphs by optimizing modularity. Graph parameters are selected based on clustering stability across resolution levels, avoiding reliance on external clustering objectives or prior assumptions about the number of modes $k$.

We augment graph edge weights with trajectory-level Jacobian features estimated through finite differences, as they showed low correlation with the embeddings (detailed in Appendix D). If the Jacobian features, which include data-driven information on control sensitivity and temporal variance sensitivity, are not redundant with the learned embeddings, we modulate edge weights by behavioral similarity, strengthening or weakening the edges between trajectories. Incorporating Jacobian similarity preserves the decoupling from reward learning, while helping separate distinct behavioral patterns. This is especially useful in embedding spaces with a continuous manifold structure, where embedding similarity alone may be insufficient to separate closely related behaviors. These statistical features summarize average local state–action sensitivities along each trajectory and aim to capture behavioral dynamics independently of reward structure or task specification.

For each discovered cluster, we independently apply a single-intention IRL algorithm (we test our framework with deep learning based methods to allow non-linear reward functions) to learn a reward function for each cluster. This decoupling allows reward learning to operate on coherent behaviors identified purely from behavioral similarity.

### 4.3 Adaptation to new behaviors

When unseen trajectories arrive, CoMI-IRL adapts through encoder finetuning followed by two-stage clustering to handle embedding drift.

#### 4.3.1 Encoder Finetuning

To incorporate unseen behaviors and limit catastrophic forgetting, we restructure the embedding space to map new trajectories while preserving the geometry induced by past data. This is done by finetuning the BE on the old and new data with the addition of a stability regularizer to limit the latent drift of the learned structure. Specifically, we adopt a *less-forget constraint* (Hou et al., 2019) loss, which preserves the geometric configuration by fixing the original embeddings and computing a novel distillation loss on the new embeddings. This regularizer implicitly implements rehearsal by revisiting previously observed trajectories during finetuning.

Formally, let $b_i$ be the L2-normalized embedding of a rehearsed sample generated by the frozen (reference) encoder, and let $a_i$ be the corresponding embedding produced by the finetuned encoder. The stability loss is defined as: $\mathcal{L}_{\text{stab}} = \frac{1}{N} \sum_{i=1}^{N} \left(1 - a_i^\top b_i\right)$. This loss encourages new embeddings to align with the orientation of the original representation by penalizing cosine dissimilarity.

#### 4.3.2 Finetuned Clustering and New Reward Learning

Despite stability regularization, finetuning can cause embedding drift that invalidates original cluster assignments computed in the pre-finetuning space. We address this with a two-stage clustering procedure (further detailed in Appendix): First, we re-cluster only the seen data leveraging the known expected cluster count $K_{\text{baseline}}$ from the baseline space. The algorithm prioritizes configurations producing exactly $K_{\text{baseline}}$ clusters, yielding recovered centroids and radii that represent baseline modes in the new embedding space. Online trajectories within a novelty threshold $\theta$ of any recovered centroid are assigned to that cluster; those

outside all radii are marked as novel candidates. Novel candidates are then sub-clustered via connected component analysis or Leiden clustering, following the same process described in Section 4.2. Recovered clusters retain their baseline agents, novel clusters trigger the training of new policies.

# 5 Experiments

In our experiments, we first analyze how the relationship between the assumed ($K$) and true ($K^*$) number of behavioral modes (i.e., $K \neq K^*, K = K^*$) impacts clustering and reward learning. Second, we visualize the embedding space and analyze the dataset as partitioned by our model. Finally, we test the adaptability of our approach to new behaviors.

Our main baseline is Ess-InfoGAIL (Fu et al., 2023), one of the current state-of-the-art reward-first coupled methods for MI-IRL which requires a small amount of labeled data and $K$ fixed a priori. While methods inferring $K$ exist, they are often not computationally feasible on continuous spaces (cf. Appendix E.1, where we report the clustering and reward learning results applying the nonparametric Bayesian IRL approach from Michini et al. (2015)). We also include baselines combining different clustering algorithms (K-Means (Lloyd, 1982; MacQueen, 1967), and our graph-based clustering) as versions of our behavior-first approach without contrastive embeddings, some requiring a fixed number of clusters $K$ and others inferring it. We test CoMI-IRL and each of the baseline methods using Generative Adversarial Imitation Learning (GAIL) (Ho & Ermon, 2016) and Adversarial IRL (AIRL) (Fu et al., 2017) as reward learning algorithms, with an exception for Ess-InfoGAIL, as it is based on GAIL and not easily transferable to AIRL. We deploy all methods on five MuJoCo environments: Reacher-v4, Pusher-v4, Walker2D-v4, Hopper-v4 and HalfCheetah-v4. For each environment (described in more detail, including objectives and setup, in the Appendix), we apply RL to pre-train $K^*$ expert policies, each corresponding to $K^*$ different behavior modes, following Fu et al. (2023), and for each policy we sample a set of 100 expert demonstrations.

We test all approaches over five seeds, reporting the clustering performance through Normalized Mutual Information (NMI) (Xu et al., 2003), Adjusted Rand Index (ARI) (Rand, 1971), Silhouette Score (Sil) (Rousseeuw, 1987), and the reward learning performance through Average Task Reward (ATR) (Fu et al., 2023). NMI quantifies the correlation between two clusterings (in this case, the ground truth and the result of the clustering phase), while ARI measures their similarity. Both assume values between 0 and 1, with a higher value indicating a better performance. Sil measures how well-separated and distinct clusters are in an unsupervised clustering analysis, and ranges from $-1$ to 1. To calculate the ATR of each method in each specific environment, we generate 100 rollouts for each behavioral mode using each of the considered methods, and we normalize all the results related to the expert policy, meaning that an ATR of 1 corresponds to the learned policy behaving as well as the expert one. We further analyze the reward learning performance by calculating the Equivalent Policy Invariant Comparison (EPIC) distance (Gleave et al., 2021) between the learned and ground truth rewards. The EPIC distance canonicalizes the reward models over a sampled coverage transition distribution, mapping them to the interval $[0, 1]$, where 0 indicates a perfect positive correlation of canonical rewards. In Appendix C, we report all the configurations and hyperparameters used for each experiment and for each method. We also report computational considerations (which show comparable running times between CoMI-IRL and Ess-InfoGAIL) in Appendix F.

## 5.1 Clustering and Reward Analysis

In this experiment, we show how the relationship between $K$ and $K^*$ impacts clustering and reward learning. Ess-InfoGAIL and KMeans-based baselines(K-GAIL/AIRL) require fixed $K$, while the Graph-based baseline (G-GAIL/AIRL) and CoMI-IRL infer it from data. As Ess-InfoGAIL requires a small subset of labeled demonstrations (1% of the unlabeled demonstrations) for each behavioral mode, in the $K<K^*$ case we are simulating a scenario in which the labeling underspecifies the number of latent behaviors. To do so, we remove the labeled trajectories for three modes out of six, but we keep the entire unlabeled dataset. For $K>K^*$ case, we duplicate the labeled trajectories, mimicking an overspecification in the number of latent behaviors.

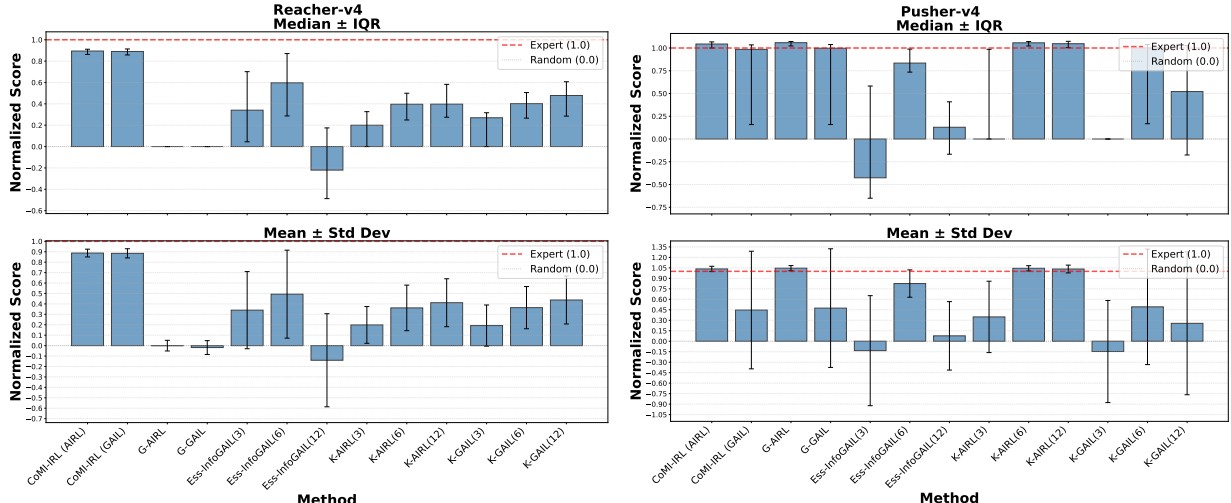

Figure 2: Median±IQR and Mean±Std for Reacher.

Figure 3: Median±IQR and Mean±Std for Pusher

In Table 1, we report the mean and standard deviation of the clustering quality metrics (NMI, ARI, Sil) for each model in each environment. In Figures 2 to 6, we report the reward learning performance in terms of ATR. For each model on each environment, we report the mean±standard deviation, and we also include the median±interquartile range (IQR), to account for potential outlier across seeds and modes and provide a more comprehensive evaluation. For all conditions, we test the models on the modes they have been trained on, either fixed or inferred.

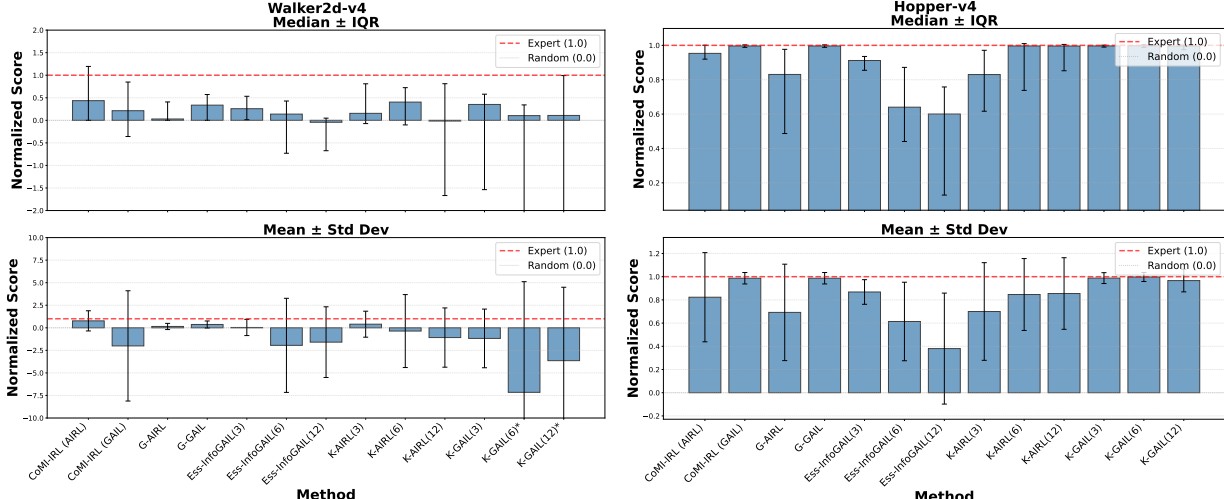

Figure 4: Median±IQR and Mean±Std for Walker2d.

Figure 5: Median±IQR and Mean±Std for Hopper.

In CoMI-IRL, AIRL and GAIL are applied to the same datasets. Thus, the performance differences between CoMI-IRL versions come from the two paradigms. The differences between CoMI-IRL(GAIL) and Ess-InfoGAIL stem from the behavior-first perspective.

The results show how, without the embeddings, decoupling can lead to high performance in case of $K=K^*$ and clearly separable behavioral modes (such as in the Pusher, Hopper and HalfCheetah cases, where K-Means with $K=K^*$ and Graph clustering have perfect NMI/ARI, and ATR comparable with CoMI-IRL), with the simple behavior-first methods outperforming Ess-InfoGAIL. When trajectories are not clearly separable in the original space (such as in the Reacher and Walker2d cases) or $K\neq K^*$, clustering performance drops. This

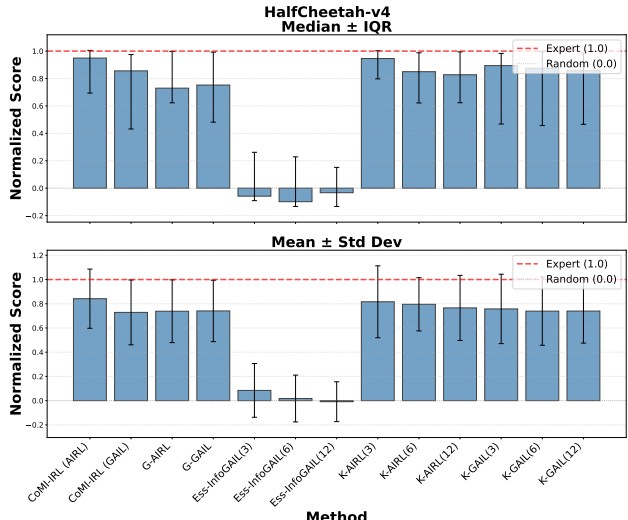

Figure 6: Median±IQR and Mean±Std for HalfCheetah.

Table 1: Clustering performance of all model categories across the considered environments. Results are reported as mean $\pm$ standard deviation over five seeds. Higher values are better for all metrics, with best results bolded. For Reacher and Pusher, $K^* = 6$, while for Walker, Hopper and HalfCheetah $K^* = 3$.

| Metric | Model | Reacher | Pusher | Walker | Hopper | HalfCheetah |
|--------|-------|---------|--------|--------|--------|-------------|
| NMI ↑ | (K=3)-GAIL/AIRL | $0.16 \pm 0.02$ | $0.72 \pm 0.00$ | $0.53 \pm 0.02$ | $\mathbf{1.00 \pm 0.00}$ | $0.71 \pm 0.06$ |
| | (K=6)-GAIL/AIRL | $0.43 \pm 0.04$ | $\mathbf{1.00 \pm 0.00}$ | $0.45 \pm 0.03$ | $0.76 \pm 0.00$ | $0.73 \pm 0.00$ |
| | (K=12)-GAIL/AIRL | $0.41 \pm 0.01$ | $0.87 \pm 0.01$ | $0.45 \pm 0.02$ | $0.66 \pm 0.01$ | $0.68 \pm 0.02$ |
| | G-GAIL/AIRL | $0.01 \pm 0.00$ | $\mathbf{1.00 \pm 0.00}$ | $0.50 \pm 0.00$ | $\mathbf{1.00 \pm 0.00}$ | $0.90 \pm 0.00$ |
| | (K=3) Ess-InfoGAIL | $0.53 \pm 0.03$ | $0.74 \pm 0.06$ | $0.45 \pm 0.07$ | $\mathbf{1.00 \pm 0.00}$ | $0.76 \pm 0.06$ |
| | (K=6) Ess-InfoGAIL | $0.68 \pm 0.03$ | $0.90 \pm 0.04$ | $0.36 \pm 0.08$ | $0.58 \pm 0.01$ | $0.62 \pm 0.06$ |
| | (K=12) Ess-InfoGAIL | $0.54 \pm 0.03$ | $0.62 \pm 0.07$ | $0.34 \pm 0.04$ | $0.42 \pm 0.04$ | $0.35 \pm 0.11$ |
| | **CoMI-IRL (Ours)** | $\mathbf{0.84 \pm 0.00}$ | $\mathbf{1.00 \pm 0.00}$ | $\mathbf{0.78 \pm 0.04}$ | $\mathbf{1.00 \pm 0.00}$ | $\mathbf{1.00 \pm 0.00}$ |
| ARI ↑ | (K=3)-GAIL/AIRL | $0.08 \pm 0.01$ | $0.48 \pm 0.00$ | $0.48 \pm 0.05$ | $\mathbf{1.00 \pm 0.00}$ | $0.66 \pm 0.10$ |
| | (K=6)-GAIL/AIRL | $0.26 \pm 0.02$ | $\mathbf{1.00 \pm 0.00}$ | $0.33 \pm 0.05$ | $0.58 \pm 0.01$ | $0.66 \pm 0.00$ |
| | (K=12)-GAIL/AIRL | $0.21 \pm 0.02$ | $0.75 \pm 0.05$ | $0.30 \pm 0.03$ | $0.41 \pm 0.01$ | $0.51 \pm 0.08$ |
| | G-GAIL/AIRL | $0.00 \pm 0.00$ | $\mathbf{1.00 \pm 0.00}$ | $0.45 \pm 0.00$ | $\mathbf{1.00 \pm 0.00}$ | $0.93 \pm 0.00$ |
| | (K=3) Ess-InfoGAIL | $0.45 \pm 0.03$ | $0.69 \pm 0.08$ | $0.40 \pm 0.09$ | $\mathbf{1.00 \pm 0.00}$ | $0.67 \pm 0.13$ |
| | (K=6) Ess-InfoGAIL | $0.57 \pm 0.05$ | $0.78 \pm 0.05$ | $0.25 \pm 0.07$ | $0.36 \pm 0.04$ | $0.45 \pm 0.06$ |
| | (K=12) Ess-InfoGAIL | $0.30 \pm 0.04$ | $0.33 \pm 0.06$ | $0.15 \pm 0.04$ | $0.13 \pm 0.03$ | $0.17 \pm 0.12$ |
| | **CoMI-IRL (Ours)** | $\mathbf{0.63 \pm 0.00}$ | $\mathbf{1.00 \pm 0.00}$ | $\mathbf{0.81 \pm 0.05}$ | $\mathbf{1.00 \pm 0.00}$ | $\mathbf{1.00 \pm 0.00}$ |
| Sil ↑ | (K=3)-GAIL/AIRL | $0.09 \pm 0.00$ | $0.16 \pm 0.00$ | $0.06 \pm 0.02$ | $0.63 \pm 0.00$ | $0.31 \pm 0.00$ |
| | (K=6)-GAIL/AIRL | $0.08 \pm 0.00$ | $0.15 \pm 0.00$ | $0.04 \pm 0.01$ | $0.09 \pm 0.00$ | $0.41 \pm 0.00$ |
| | (K=12)-GAIL/AIRL | $0.08 \pm 0.00$ | $0.06 \pm 0.01$ | $0.02 \pm 0.02$ | $0.20 \pm 0.01$ | $0.33 \pm 0.06$ |
| | G-GAIL/AIRL | $0.12 \pm 0.00$ | $0.15 \pm 0.00$ | $0.10 \pm 0.00$ | $0.63 \pm 0.00$ | $0.30 \pm 0.00$ |
| | (K=3) Ess-InfoGAIL | $0.11 \pm 0.03$ | $0.37 \pm 0.08$ | $0.03 \pm 0.01$ | $0.18 \pm 0.02$ | $0.26 \pm 0.16$ |
| | (K=6) Ess-InfoGAIL | $0.03 \pm 0.01$ | $0.41 \pm 0.06$ | $0.00 \pm 0.02$ | $0.03 \pm 0.14$ | $0.22 \pm 0.09$ |
| | (K=12) Ess-InfoGAIL | $-0.06 \pm 0.03$ | $0.17 \pm 0.04$ | $-0.06 \pm 0.04$ | $0.30 \pm 0.18$ | $0.22 \pm 0.07$ |
| | **CoMI-IRL (Ours)** | $\mathbf{0.87 \pm 0.05}$ | $\mathbf{0.98 \pm 0.01}$ | $\mathbf{0.59 \pm 0.02}$ | $\mathbf{0.91 \pm 0.08}$ | $\mathbf{0.88 \pm 0.06}$ |

degrades the reward learning, causing mode collapse in certain Walker cases, denoted by high variance and a negative ATR. CoMI-IRL, by working on a restructured space, outperforms the baselines both in clustering and reward learning. Ess-InfoGAIL fails on HalfCheetah, likely due to the complexity of the environment. In the clearly separable cases (Pusher and Hopper), it is clear the connection between the clustering quality

and ATR performance for Ess-InfoGAIL. In both environments, Ess-InfoGAIL manages to achieve good performance only when $K=K^*$, with a clear decline (in Hopper) or collapse (in Pusher) when $K \neq K^*$.

In the $K<K^*$ case, Ess-InfoGAIL is forced to explain all $K^*$ behavioral modes with $K$ latent codes, meaning that trajectories from other behavioral modes in the dataset will contaminate the training of the $K$ considered classes, as it shows with a low NMI and ATR. In the $K>K^*$ case, it is forced to use all latent codes even if the dataset does not require all the modes, also leading to a lower NMI.

We test the statistical significance of the performance through a Wilcoxon signed-rank test (Wilcoxon, 1945) with Benjamini-Hochberg False Discovery Rate (FDR) correction ($\alpha = 0.05$) (Benjamini & Hochberg, 1995) and Cohen's $d_z$ for the paired effect size (Cohen, 2013). The results, presented for each environment in Appendix G, provides evidence supporting the observed ATR performance. CoMI-IRL demonstrates significant advantages over Ess-InfoGAIL in all environments (minimum $p_{\text{FDR}} < 6.5 \times 10^{-15}$, effect sizes $d_z = 1.07$–$4.65$ across environments). The Wilcoxon signed-rank tests with FDR correction indicate statistically significant differences across all environments, and the effect sizes suggest that these differences are large in magnitude. Overall, this is consistent with the hypothesis that a decoupled behavior-first paradigm offers advantages over a reward-first coupled one in these evaluated tasks. CoMI-IRL finding $K=12$ classes suggests that the $K^*$ pre-trained expert policies actually produced a finer-grained behavioral structure with twelve different behavioral modes, and we investigate this further.

### 5.2 Behavior Visualization

In Figure 7, for each environment, we visualize the original space of trajectories and the respective embedding space built by the BE. For both, we use UMAP (McInnes et al., 2018) as it preserves more of the global structure of the space. By visualizing the embeddings produced by the BE, it is possible to analyze how behaviors relate to each other without introducing any assumption in terms of reward.

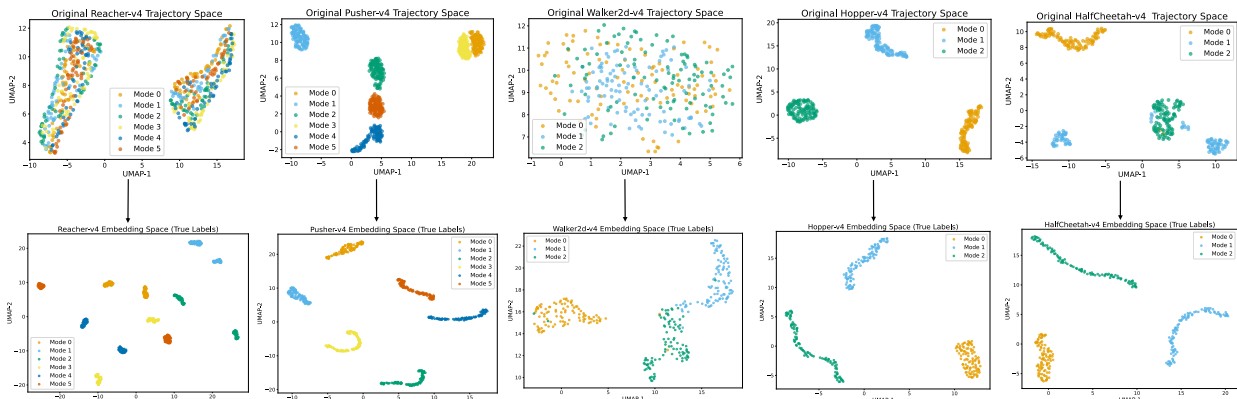

Figure 7: 2D UMAP visualizations of the original trajectory space at the top of each environment (from left to right: Reacher, Pusher, Walker2D, Hopper, HalfCheetah), and the respective embedding space resulting from the BE at the bottom.

As visible in Figure 7, the trajectories of Pusher and Hopper are already fairly separable in the original space, explaining K-GAIL and G-GAIL perfect clustering performance when $K = K^*$. Ess-InfoGAIL approaches the task following the reward-first paradigm and does not consider the inherit similarity between trajectories of the original space. As a result, Ess-InfoGAIL does not manage to perfectly disentangle the modes when the original space is not already separable, highlighting an advantage of the decoupling.

In Reacher, our model notably detects double the amount of true behavioral clusters (twelve instead of six). To further investigate this finer-grained behavioral structure, we analyzed the angular composition of the dataset as partitioned by our method. This analysis revealed that our method successfully disentangled the behavioral modes and discovered sub-modes, creating clusters that are consistent with respect to the configuration of the elbow joint angle (positive or negative), shown in Table 2. The behavior-first perspective

discovers what the data actually contains rather than confirming what was assumed, highlighting a concrete advantage of the behavior-first perspective over reward-first methods.

We further analyze this case and evaluate how the trained policies perform in terms of the elbow angle. We generate 50 rollouts for each agent and we analyze the angle consistency. The results are reported in Table 3. CoMI-IRL achieves low variance across all modes, consistently identifying one "Up" and one "Down" agent per mode, allowing for controllability. Ess-InfoGAIL exhibits high variance for Mode 1 ($\pm0.52$ rad), indicating that the latent code fails to capture a consistent physical strategy across seeds. In the other modes the model shows consistency, however this also means that it does not allow for selection of a preferred elbow configuration, collapsing into one configuration per mode that cannot be selected or controlled.

Looking at the MI-IRL task from a behavior-first perspective offers a level of nuance interpretability and controllability not captured by the expert definition of $K$, and that entangled models such as Ess-InfoGAIL don't permit due to the latent code structuring that reduces their flexibility. Ablation experiments over the loss components (see Appendix E.3) support DIM (cf. Section 3) as a regularization term that improves stability and performance, and ablation experiments over the architecture confirm the pre-processing block ($RFF + MLP + CNN$) as a crucial component on overlapping spaces, highlighting the segment and pairwise losses as key loss components. We also studied the effect of the Jacobian reweighting (see Appendix E.4, which showed a small impact in well-conditioned learned spaces (such as Reacher or Pusher), where the number of increased edges is between 1% and 5%. On Walker, which has more heterogeneous action effects, it leads to a decrease of edges between clusters, and a 13% increase of within-cluster edges.

Table 2: **Disentanglement of elbow angle.** CoMI-IRL splits each target mode into two distinct strategy clusters: "Elbow-Down" ($\theta < 0$) and "Elbow-Up" ($\theta > 0$). Values are in radians (Mean $\pm$ SD).

| Mode | Elbow-Down | | Elbow-Up | |
|------|-----|-----|-----|-----|
| | ID | $\theta$ | ID | $\theta$ |
| 0 | C8 | $-0.92 \pm 0.14$ | C4 | $+1.96 \pm 0.30$ |
| 1 | C7 | $-0.77 \pm 0.19$ | C5 | $+1.30 \pm 0.24$ |
| 2 | C10 | $-1.18 \pm 0.31$ | C6 | $+0.80 \pm 0.19$ |
| 3 | C11 | $-1.99 \pm 0.24$ | C2 | $+0.90 \pm 0.20$ |
| 4 | C9 | $-2.08 \pm 0.18$ | C1 | $+1.62 \pm 0.24$ |
| 5 | C0 | $-1.63 \pm 0.20$ | C3 | $+2.16 \pm 0.22$ |

Table 3: **Strategy Consistency.** Comparison of learned elbow angles ($\theta$ in radians, Mean $\pm$ SD). CoMI-IRL consistently identifies two distinct strategies per mode, whereas Ess-InfoGAIL collapses them, leading to high variance (e.g., Mode 1).

| | CoMI-IRL (Ours) | | Ess-InfoGAIL |
|------|-----|-----|-----|
| Mode | Down ($\theta$) | Up ($\theta$) | Angle ($\theta$) |
| 0 | $-0.85 \pm 0.02$ | $+1.79 \pm 0.04$ | $+1.90 \pm 0.03$ |
| 1 | $-0.71 \pm 0.03$ | $+1.15 \pm 0.03$ | $+\mathbf{0.21} \pm \mathbf{0.52}$ |
| 2 | $-1.09 \pm 0.01$ | $+0.71 \pm 0.01$ | $-0.18 \pm 0.18$ |
| 3 | $-1.82 \pm 0.05$ | $+0.84 \pm 0.02$ | $-1.86 \pm 0.06$ |
| 4 | $-2.02 \pm 0.04$ | $+1.51 \pm 0.07$ | $-1.89 \pm 0.17$ |
| 5 | $-1.54 \pm 0.06$ | $+1.95 \pm 0.08$ | $+1.94 \pm 0.04$ |

## 5.3 Reward distance analysis

To further analyze reward learning beyond the policy rollout performance, we calculate the EPIC (Gleave et al., 2021) distance between the learned rewards and the ground truth rewards. For each environment, we compare all baselines against the ground truth, and we calculate the statistical significance by comparing with the best-performing CoMI-IRL configuration (in terms of lowest mean distance, highlighted in bold). The full results are reported in Appendix E.2.

In Table 4, we report the EPIC distance for Pusher-v4, Hopper-v4, HalfCheetah-v4, and Walker2d-v4, between CoMI-IRL and Ess-InfoGAIL, with CoMI-IRL showing the highest reward alignment across all environments. Table 5 evaluates the reward learning in the Reacher-v4 task, showing the effect of sub-mode disentanglement. Configuration-sensitive reward functions, such as those learned in CoMI-IRL's pure sub-clusters, naturally penalize alternative configurations. Because the ground-truth reward only evaluates fingertip distance and is configuration-indifferent, single-mode evaluation distances are high. However, by taking the max-reduction over the sub-mode reward networks (the Combined rows), the EPIC distance

Table 4: EPIC distance (Mean $\pm$ Std) to the true reward function. Comparisons and exact $p$-values are computed against the best-performing CoMI-IRL configuration (in bold) for each environment. Statistically non-significant differences ($p \geq 0.05$) are marked with [n.s.].

| Method | Pusher-v4 | Hopper-v4 | HalfCheetah-v5 | Walker2d-v4 |
|---|---|---|---|---|
| CoMI-IRL (AIRL) | $\mathbf{0.3360 \pm 0.1081}$ | $\mathbf{0.5639 \pm 0.1691}$ | $0.4409 \pm 0.0478$ | $0.6134 \pm 0.0895$ |
| | (Reference) | (Reference) | ($p = 0.0059$) | ($p = 0.2676$ [n.s.]) |
| CoMI-IRL (GAIL) | $0.4938 \pm 0.0884$ | $0.7159 \pm 0.1445$ | $\mathbf{0.3746 \pm 0.0673}$ | $\mathbf{0.5759 \pm 0.0919}$ |
| | ($p = 1.12 \times 10^{-7}$) | ($p = 0.0164$) | (Reference) | (Reference) |
| Ess-InfoGAIL (K=12) | $0.6528 \pm 0.0888$ | $0.7037 \pm 0.0848$ | $0.7928 \pm 0.0950$ | $0.7338 \pm 0.0773$ |
| | ($p = 2.60 \times 10^{-18}$) | ($p = 0.0082$) | ($p = 1.20 \times 10^{-15}$) | ($p = 4.79 \times 10^{-6}$) |
| Ess-InfoGAIL (K=3) | $0.7603 \pm 0.1037$ | $0.7024 \pm 0.1737$ | $0.6178 \pm 0.0854$ | $0.6785 \pm 0.0449$ |
| | ($p = 4.70 \times 10^{-13}$) | ($p = 0.0187$) | ($p = 7.37 \times 10^{-12}$) | ($p = 0.0009$) |
| Ess-InfoGAIL (K=6) | $0.6415 \pm 0.0702$ | $0.6848 \pm 0.1478$ | $0.7011 \pm 0.1532$ | $0.6608 \pm 0.0575$ |
| | ($p = 2.60 \times 10^{-17}$) | ($p = 0.0234$) | ($p = 9.99 \times 10^{-17}$) | ($p = 0.0037$) |

Table 5: EPIC distance (Mean $\pm$ Std) for Reacher-v4. Single models are compared against CoMI-IRL (AIRL), and combined (max-reduced) models are compared against CoMI-IRL (AIRL) (Combined).

| Method | Type | EPIC Distance | $p$-value vs. Ref |
|---|---|---|---|
| **CoMI-IRL (AIRL)** | Single | $\mathbf{0.7113 \pm 0.0274}$ | (Reference) |
| **CoMI-IRL (AIRL) (Combined)** | Combined | $\mathbf{0.5373 \pm 0.0595}$ | (Reference) |
| CoMI-IRL (GAIL) | Single | $0.7762 \pm 0.0317$ | $8.03 \times 10^{-22}$ |
| CoMI-IRL (GAIL) (Combined) | Combined | $0.6165 \pm 0.0994$ | $0.0006$ |
| Ess-InfoGAIL (K=12) | Single | $0.7060 \pm 0.1167$ | $0.7341$ [n.s.] |
| Ess-InfoGAIL (K=12) (Combined) | Combined | $0.7109 \pm 0.1217$ | $2.00 \times 10^{-8}$ |
| Ess-InfoGAIL (K=3) | Single | $0.6598 \pm 0.0478$ | $0.0013$ |
| Ess-InfoGAIL (K=6) | Single | $0.6207 \pm 0.0939$ | $1.59 \times 10^{-5}$ |

lowers. This combined reward is statistically significantly better than all other combined baseline rewards. This is consistent with the reward formulation proposed by Chen et al. (2020a; 2023), where the reward is decomposed into a task component and a strategy component. Following such formulation, the encoding of behavioral differences (the "strategies") into the specific reward functions contributes to a higher alignment with the ground truth reward.

### 5.4 Adaptation to new behaviors

In this experiment, we evaluate CoMI-IRL's adaptability to new behaviors. Entangled methods cannot adapt to unseen behaviors without full retraining. We consider the case of an initial dataset containing $K_{\text{base}} = K^*/2$ behavioral modes, with the other half received at a later moment. We hereby show how CoMI-IRL does not need to retrain the entire framework, but it is adaptable to unseen behaviors and maintains previously learned knowledge. For Reacher and Pusher, we remove half of the modes from the original dataset, apply CoMI-IRL for the remaining ones, and then finetune adding the new behaviors. For Walker-2D, Hopper and HalfCheetah, which only have three modes (forward, backwards, still), we remove the third mode. We report NMI, ARI, Silhouette and ATR (using CoMI-IRL(AIRL)) for $K_{\text{base}}$ and $K_{\text{finetuned}}$ in Table 6.

The results show how CoMI-IRL adapts to new behaviors through finetuning. In the first four environments, CoMI-IRL manages to maintain performance between stages. The finetuned model correctly identifies $K$ on Reacher and Pusher, but it oversegments on Hopper and HalfCheetah, even though performance was not significantly hurt.

On Walker-2D, NMI/ARI are high on $K_{\text{base}}$ as the first two modes (moving forward or backward) are separable. The two-stage clustering allows the model to more selectively isolate novel trajectories, leading to higher NMI/ARI and ATR on $K_{\text{finetuned}}$ compared to the full dataset experiment. In this case, the

Table 6: **Adaptation Performance.** NMI, ARI, Silhouette, and ATR scores before ($K_{\text{base}}$) and after ($K_{\text{finetuned}}$) adapting to unseen behaviors. $K^*$ denotes the total number of modes in the final dataset and $\hat{K}$ the number of inferred ones.

| Stage | NMI | ARI | Sil | ATR | $\hat{K}$ |
|---|---|---|---|---|---|
| **Reacher-v4** ($K^* = 6$) | | | | | |
| $K_{\text{base}}$ | 0.76±0.00 | 0.57±0.00 | 0.85±0.00 | 0.93±0.11 | 6±0.00 |
| $K_{\text{finetuned}}$ | 0.84±0.00 | 0.63±0.00 | 0.81±0.00 | 0.93±0.10 | 12±0.00 |
| **Pusher-v4** ($K^* = 6$) | | | | | |
| $K_{\text{base}}$ | 1.00±0.00 | 1.00±0.00 | 0.76±0.00 | 1.10±0.14 | 3±0.00 |
| $K_{\text{finetuned}}$ | 1.00±0.00 | 1.00±0.00 | 0.76±0.00 | 1.09±0.12 | 6±0.00 |
| **Hopper-v4** ($K^* = 3$) | | | | | |
| $K_{\text{base}}$ | 1.00±0.00 | 1.00±0.00 | 1.00±0.00 | 0.97±0.06 | 2±0.00 |
| $K_{\text{finetuned}}$ | 0.87±0.01 | 0.84±0.01 | 0.86±0.03 | 0.93±0.05 | 5±0.00 |
| **HalfCheetah-v4** ($K^* = 3$) | | | | | |
| $K_{\text{base}}$ | 1.00±0.00 | 1.00±0.00 | 1.00±0.00 | 0.96±0.13 | 2±0.00 |
| $K_{\text{finetuned}}$ | 0.90±0.02 | 0.87±0.03 | 0.86±0.03 | 0.91±0.10 | 4±0.00 |
| **Walker2D-v4** ($K^* = 3$) | | | | | |
| $K_{\text{base}}$ | 0.98±0.02 | 0.99±0.01 | 0.54±0.05 | 0.57±0.31 | 2±0.00 |
| $K_{\text{finetuned}}$ | 0.90±0.01 | 0.88±0.01 | 0.25±0.04 | 0.87±0.62 | 4±0.00 |

oversegmentation of the new mode leads to an unbalanced set of clusters, dividing the new trajectories into a big and a small cluster. The performance of the agent trained on the small cluster leads to higher standard deviation, while the performance of the agent trained on the big cluster raise the mean ATR.

## 6 Limitations & Future Work

While results are encouraging, there are limitations. CoMI-IRL suffers from known Transformers limitations (Sanford et al., 2023), relying on the learned space quality and hyperparameter choice. Nonetheless, visualizing the embedding space allows for a qualitative analysis of the results before the IRL phase. The clustering algorithm also plays a determinant role, and our method automatically finds a stable structure for the space considered. A deeper analysis of the Jacobian edge reweighting hyperparameters is part of future studies. While CoMI-IRL has been shown to adapt, future work includes a deeper study on the scalability to a large scale of multiple modes, and the robustness to catastrophic forgetting. Scalability costs can increase when considering a high number of clusters, but as all reward learning processes are independent parallelization is highly impactful. The stability regularizer with rehearsal mechanism and the structural isolation given by the decoupling suggest a good degree of robustness, but further work is needed. This includes an analysis of the relationship between learned rewards to potentially individuate oversegmentation cases. Other future directions include a higher focus on the reward learning phase, taking into account the connection between clusters representing different ways of achieving the same task. Finally, interpretability results need to be assessed on a user study with real-world data, as real-world scenarios might have more subtle differences, with a focus on understanding the fine-grained distinctions and their relevance. This could also be explored in combination with potentially introducing human feedback in the process, to study the alignment between demonstrated behavioral modes and stated intentions.

## 7 Conclusions

In this work, we highlighted how the current literature approaches the MI-IRL problem following a reward-first paradigm: by asking which reward function, if followed, would most likely have generated each trajectory in a dataset, without considering the intrinsic relationships between behaviors. We presented CoMI-IRL, a

transformer-based unsupervised framework for MI-IRL which follows a behavior-first paradigm and decouples behavior clustering from reward learning: it first learns trajectory embeddings, clusters them, and then applies single-reward IRL to each discovered cluster. In our experiments on the MuJoCo benchmark, CoMI-IRL produced better clustering performance and downstream policy performance compared to the baseline. We analyzed how the relationship between $K$ and $K^*$ impacts existing methods, while our approach infers $K$ from data and shows adaptability to unseen behaviors. Decoupling also allowed us to analyze the dataset as partitioned by our model in terms of behavioral similarity and not through the learned rewards, facilitating the interpretation of similarities in the dataset. Beyond the empirical improvements, CoMI-IRL represents a shift from a reward-first paradigm, in which behavioral structure is discovered through reward-generating likelihood, to a behavior-first paradigm, in which reward learning follows from behavioral structure discovered through dynamic similarity. This shift is conceptually prior to the specific architectural choices made here, and opens directions for MI-IRL research such as unsupervised behavioral analysis, sub-mode discovery, and adaptation to new behaviors without reward re-specification.

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

## Appendix

In this document, we provide further material regarding the CoMI-IRL paper, including details on the environments (Section A), the Behavioral Encoder architecture (Section B), hyperparameters C and the graph-based clustering algorithm (Section D). We provide further results in Section E, including an ablation study on the architecture and loss components introduced in the main paper, and further considerations on the effect of the Jacobian Reweighting. Finally, we report the average wallclock time of CoMI-IRL and Ess-InfoGAIL for the considered environments.

## A  Environments

Reacher consists a 2-DoF robot arm which goal is to move the robot's end effector close to a target position. Pusher consists a 4-DoF robot arm which goal is to move a target cylinder to a goal position using the robot's end effector. Both environments have six goals and the goal location has been removed from the observation Fu et al. (2023). Walker-2D is a 6-DoF bipedal robot consisting of two legs and feet attached to a common base which goal is to make coordinate both sets of legs and feet to move the robot in the right direction, including running forward, running backward and balancing. HalfCheetah is a 6-DoF 2D robot which goal is to apply torque to the joints to make the cheetah run in the right direction (forward or backward) or balancing. Hopper is a 3-DoF one-legged robot which goal is to make hops that move the robot either forward, backward, or stay balanced. An illustration of each environment is provided in Figure 8.

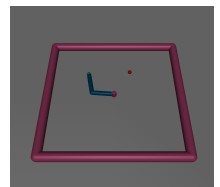 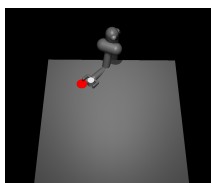 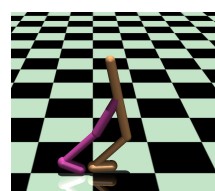 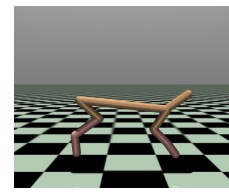 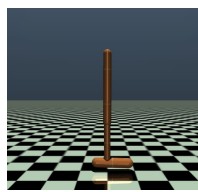

Figure 8: **Left to Right:** Reacher, Pusher, Walker2d, HalfCheetah, Hopper.

## B  Additional Architecture Details

We presented the architecture of the Behavioral Encoder (BE) in Section 3.1 of the main paper. As trajectories might contain high-frequency elements, and Multi-Layer Perceptrons ($MLPs$) knowingly struggle with high-frequency functions Rahaman et al. (2019), we use Random Fourier Features with Gaussian mapping ($RFF$) Rahimi & Recht (2007); Xu et al. (2019); Zheng et al. (2022); Li et al. (2021). In each head, this mapping injects a rich set of sinusoidal features at multiple random orientation and frequencies to overcome known $MLP$ spectral bias towards low-frequency functions so that, for each $s_{\tau_i}^j \in S_{\tau_i}$ and $a_{\tau_i}^j \in A_{\tau_i}$,

$$RFF(s_{\tau_i}^j) = [\sin(2\pi s_{\tau_i}^j W), \cos(2\pi s_{\tau_i}^j W)]^T, W \sim \mathcal{N}(0, \sigma^2), \tag{6}$$

with $\sigma$ chosen as a hyperparameter. The output is subsequently processed by a shallow $MLP$, and then followed by a 1D ($CNN$) to capture short-range temporal patterns within the behavioral trajectory, ensuring local coherency. Our approach benefits from this technique both for the internal MLPs in the Transformer, and because $RFF$ can be seen as a generalized case of the sinusoidal positional encoding Tancik et al. (2020) needed by the transformer to have information on the order of the sequence.

## C  Implementation Details

In this section, we report the hyperparameters used in our experiments for AIRL and GAIL (Table 7), for Ess-InfoGAIL (Table 8), and for the encoder (Table 9). The agents are learned using the imitation library Gleave et al. (2022) and Stable-Baselines3 Raffin et al. (2021), while we use the original repository for Ess-InfoGAIL. For Ess-InfoGAIL, we used the hyperparameter configuration for each environment as

presented by the authors of the original work. For the other methods, hyperparameters tuning has been carried out manually.

| AIRL Hyperparameters |
| --- |
| Buffer: batch=256, capacity=2048 |
| Disc: updates=32, reward_hid=(32,), pot_hid=(32,32) |
| PPO: lr=3e-4, batch=64, ent=0.01, $\gamma$=0.99 |
| clip=0.2, epochs=10 |
| Training: 250k timesteps/cluster (1.5M for Walker2d-v4) |

Table 7: AIRL configuration. Disc=discriminator, pot_hid=potential hidden sizes.

| Hyperparameter | Value | Hyperparameter | Value |
| --- | --- | --- | --- |
| use_obs_norm | True | lr_actor | 3e-3 |
| auto_lr | True | lr_critic | 3e-3 |
| rollout_length | 5000 | lr_prior | 3e-3 |
| batch_size | 1000 | lr_disc | 5e-3 |
| num_steps | 2000000 | lr_q | 1e-2 |
| eval_interval | 200000 | epoch_ppo | 20 |
| obs_horizon | 8 | epoch_disc | 50 |
| obs_his_steps | 1 | epoch_prior | 1 |
| im_ratio | 1 | surrogate_loss_coef | 4.0 |
| disc_grad_penalty | 0.1 | value_loss_coef | 5.0 |
| disc_coef | 20 | us_coef | 1.0 |
| ss_coef | 4.0 | info_max_coef1 | 3.0 |
| info_max_coef2 | 0.05 | info_max_coef3 | 0.5 |
| begin_weight | 20 | reward_i_coef | 1.0 |
| reward_us_coef | 0.1 | reward_ss_coef | 0.1 |
| reward_t_coef | 0.005 | | |

Table 8: Hyperparameters used for Ess-InfoGAIL.

| Param | Reach | Push | Walk | Hopp | HC |
| --- | --- | --- | --- | --- | --- |
| Shared: emb=32, heads=4, layers=2, hid=1024 | | | | | |
| drop=0.1, $m_s$=64, $m_a$=32 | | | | | |
| Loss: $\alpha/\beta/\gamma/\delta$=0.5/1.0/0.5/1.0; epochs=100 | | | | | |
| lr | 1e-4 | 1e-4 | 5e-4 | 1e-4 | 1e-4 |
| inp_ch | 6 | 20 | 17 | 11 | 17 |
| max_len | 50 | 100 | 100 | 100 | 100 |
| batch | 64 | 32 | 64 | 64 | 64 |
| $\sigma_s$ | 0.01 | 0.001 | 0.001 | 0.01 | 0.01 |
| $\sigma_a$ | 0.1 | 0.001 | 0.001 | 0.01 | 0.01 |

Table 9: BE hyperparameters. Reach=Reacher-v4, Push=Pusher-v4, Walk=Walker2d-v4, Hopp=Hopper-v4, HC=HalfCheetah-v4. Shared parameters use abbreviated names: emb=embedding dim, heads=attention heads, hid=hidden dim, drop=dropout, $m_s/m_a$=Gaussian means (state/action), $\sigma_s/\sigma_a$=Gaussian sigmas.

# D  Graph-Based Clustering Description

This appendix provides detailed descriptions of the graph-based clustering methods used in CoMI-IRL: the joint $k$-resolution sweep for the main clustering and the two-stage target-aware clustering for post-finetuning adaptation.

## D.1  Joint $k$-Resolution Sweep

After training the behavior encoder, we cluster trajectory embeddings using graph-based community detection. Our approach operates on the graph structure of the embedding space, making it suitable for continuous behavioral manifolds where modes may not have clear density gaps.

**Graph Construction**   Given $N$ trajectory, we take the L2-normalized embeddings $\{z_i\}_{i=1}^N$ and we construct a weighted $k$-nearest neighbor graph $\mathcal{G} = (V, E, W)$. For each node $i$, we find its $k$ nearest neighbors using cosine distance and assign RBF kernel weights based on cosine similarity:

$$w_{ij} = \exp\left(\frac{\text{sim}(z_i, z_j)}{\sigma}\right), \quad \text{sim}(z_i, z_j) = 1 - d_{\cos}(z_i, z_j) \tag{7}$$

where $d_{\cos}(z_i, z_j) = 1 - \tilde{z}_i^\top \tilde{z}_j$ and $\sigma = 1.0$. We make the graph undirected by taking the maximum weight: $w_{ij} = \max(w_{ij}, w_{ji})$.

**Automatic Structure Detection**   Before applying community detection, we check if the graph naturally decomposes into isolated components. For $k \in \{15, 30, 50, 75\}$, we compute the connected components of the $k$-NN graph. If multiple significant components exist (size $\geq$ `min_cluster_size`), use them directly as clusters, otherwise proceed to the joint sweep. This handles environments where modes form distinct, well-separated islands in the embedding space (e.g., Reacher-v4, Pusher-v4).

**Joint $k$-Resolution Sweep**   For clusters spread out on a continuous manifold (e.g., Walker2d-v4), we jointly optimize over the graph connectivity parameter $k$ and the Leiden resolution parameter $\gamma$, following Algorithm 1.

---

**Algorithm 1** Joint $k$-Resolution Sweep

---

**Require:** Embeddings $Z$, range $[k_{\min}, k_{\max}]$, resolution set $\Gamma$, min cluster size $m$
**Ensure:** Cluster labels, optimal $(k^*, \gamma^*)$
 1: $\mathcal{R} \leftarrow \emptyset$ {Results collection}
 2: **for** $k \in \text{linspace}(k_{\min}, k_{\max}, n_k)$ **do**
 3:     Build $k$-NN graph $\mathcal{G}_k$
 4:     **for** $\gamma \in \Gamma$ **do**
 5:         $\ell_{k,\gamma} \leftarrow$ Leiden clustering with resolution $\gamma$
 6:         Filter clusters with size $< m$ as noise
 7:         $\mathcal{R} \leftarrow \mathcal{R} \cup \{(k, \gamma, \ell_{k,\gamma})\}$
 8:     **end for**
 9: **end for**
10: **Stability-based Selection:**
11: **for** each $(k, \gamma) \in \mathcal{R}$ **do**
12:     $\mathcal{N}_{k,\gamma} \leftarrow$ neighbors with $|k' - k| \leq 15$ or $|\gamma' - \gamma| \leq 0.3$
13:     $\text{stability}_{k,\gamma} \leftarrow \frac{1}{|\mathcal{N}_{k,\gamma}|} \sum_{(k',\gamma') \in \mathcal{N}_{k,\gamma}} \text{ARI}(\ell_{k,\gamma}, \ell_{k',\gamma'})$
14: **end for**
15: $(k^*, \gamma^*) \leftarrow \arg\max_{k,\gamma} \text{stability}_{k,\gamma}$
16: **return** $\ell_{k^*,\gamma^*}, (k^*, \gamma^*)$

---

The resolution set used is $\Gamma = \{0.01, 0.025, 0.05, 0.1, 0.15, 0.2, 0.25, 0.3, 0.5, 0.7, 1.0, 1.5, 2.0\}$. Of the tested environments, Pusher and Reacher did not require the joint sweep as their structure already decomposed into isolated components. Further details are provided in Section F.

**Leiden Community Detection**   The Leiden algorithm Traag et al. (2018) optimizes modularity with a resolution parameter:

$$Q_\gamma = \frac{1}{2m} \sum_{ij} \left[ A_{ij} - \gamma \frac{k_i k_j}{2m} \right] \delta(c_i, c_j) \tag{8}$$

where $A_{ij}$ is the adjacency weight, $k_i = \sum_j A_{ij}$ is the weighted degree, $m = \frac{1}{2} \sum_{ij} A_{ij}$ is the total edge weight, $c_i$ is the community of node $i$, and $\gamma$ controls granularity (higher values yield more clusters).

Rather than selecting by a single metric, we use partition stability as the primary criterion. For each $(k, \gamma)$ configuration, we compute the average Adjusted Rand Index (ARI) between its partition and those of neighboring configurations in parameter space. Configurations whose partitions are robust to small parameter perturbations receive higher stability scores. This approach avoids over-fitting to specific parameter choices and identifies genuine cluster structure.

**Jacobian-Based Edge Reweighting**   For environments with continuous behavioral manifolds, embedding similarity alone may not separate subtly distinct modes. In all environments, we augment edge weights with Jacobian-based behavioral features. For each trajectory, we compute:

- State differences: $\Delta s_t = s_{t+1} - s_t$

- Control sensitivity: $\gamma_t = \|\Delta s_t\|/(\|a_t\| + \epsilon)$

- Covariance-based features from $\text{Cov}(\Delta s, a)$

The resulting 8-dimensional feature vector captures:

- Mean/std/max of control sensitivity

- Temporal variance of sensitivity

- Top singular value and ratio from state-action covariance

- Mean/std of action magnitude

Given these statistical features $\{b_i\}$, we update edge weights:

$$w_{ij}^{\text{new}} = w_{ij} \cdot [1 + \alpha \cdot (2 \cdot b_{ij} - 1)] \tag{9}$$

where $w_{ij}^{\text{behav}} = \exp(-\|b_i - b_j\|^2/2\sigma_b^2)$ and $\alpha \in [0, 1]$ controls the behavioral contribution, which we set at 0.3. We modulate edge weights by behavioral similarity: edges between trajectories with similar dynamics $b_{ij}{>}0.5$ are strengthened, while edges between behaviorally dissimilar trajectories $b_{ij}{<}0.5$ are weakened.

Before deciding to use Jacobian features, we test if they are redundant with the learned embeddings by computing the Pearson and Spearman correlation between embedding-based and Jacobian-based pairwise similarities. If the average correlation exceeds 0.7, Jacobian features are skipped as the encoder already captures the relevant dynamics. However, all tests showed an average correlation lower than 0.45, meaning that the Jacobian features are capturing different information.

After clustering, we build a registry storing per-cluster metadata, which is used for novel detection:

$$\text{Registry}[k] = \left\{ \begin{array}{l} \mu_k = \frac{1}{|C_k|} \sum_{i \in C_k} \tilde{z}_i \quad \text{(L2-normalized centroid)} \\ r_k^{95} = \text{quantile}_{95}\left(\{d_{\cos}(\tilde{z}_i, \mu_k)\}_{i \in C_k}\right) \quad \text{(radius)} \\ n_k = |C_k| \quad \text{(count)} \end{array} \right\} \tag{10}$$

### D.2 Two-Stage Clustering Details

After finetuning the encoder on combined seen and online data, the embedding space shifts due to adaptation. This invalidates baseline cluster labels computed in the pre-finetuning space. We address this with a two-stage procedure that leverages knowledge of the expected baseline structure.

Let $f_\theta$ denote the baseline encoder and $f_{\theta'}$ the finetuned encoder. For a seen trajectory $\tau$, we can denote the baseline embedding as $z = f_\theta(\tau)$ and the finetuned embedding as $z' = f_{\theta'}(\tau)$.

Despite stability regularization, $z \neq z'$. More critically, the relative positions of clusters may change. A key factor is that we know the expected number of clusters $K_{\text{baseline}}$ from before the finetuning. Even though embeddings have drifted, the relative separability of seen modes is typically preserved. We re-cluster *only the seen data* $Z_{\text{seen}} = \{z'_i\}_{i=1}^{N_{\text{seen}}}$ using target-aware selection, following Algorithm 2.

---

**Algorithm 2** Stage 1: Target-Aware Recovery

---

**Require:** Finetuned seen embeddings $Z_{\text{seen}}$, target $K_{\text{baseline}}$
**Ensure:** Recovered labels, centroids, radii
 1: **// Check for isolated components first**
 2: **for** $k \in \{15, 30, 50, 75\}$ **do**
 3:    Build $k$-NN graph, compute connected components
 4:    Count significant components (size $\geq m$)
 5:    **if** n_significant $= K_{\text{baseline}}$ **then**
 6:       Use components as recovered clusters
 7:       **return**
 8:    **end if**
 9: **end for**
10: **// Target-aware $(k, \gamma)$ sweep**
11: $\mathcal{R} \leftarrow \emptyset$, $\mathcal{E} \leftarrow \emptyset$ {All results, exact matches}
12: **for** $(k, \gamma) \in$ SearchGrid **do**
13:    Run Leiden, filter small clusters
14:    $n_c \leftarrow$ number of valid clusters
15:    Compute silhouette $s$
16:    $\mathcal{R} \leftarrow \mathcal{R} \cup \{(k, \gamma, n_c, s, \text{labels})\}$
17:    **if** $n_c = K_{\text{baseline}}$ **then**
18:       $\mathcal{E} \leftarrow \mathcal{E} \cup \{|\mathcal{R}| - 1\}$
19:    **end if**
20: **end for**
21: **// Selection**
22: **if** $\mathcal{E} \neq \emptyset$ **then**
23:    Select from $\mathcal{E}$ with best silhouette
24: **else**
25:    Score $= -2 \cdot |n_c - K_{\text{baseline}}| + s$
26:    Select configuration with best score
27: **end if**
28: Compute recovered centroids $\{\mu_k^{\text{rec}}\}$ and radii $\{r_k^{\text{rec}}\}$
29: **return** labels, centroids, radii

---

The scoring function $-2 \cdot |n_c - K| + s$ heavily penalizes deviation from the target cluster count while still rewarding cluster quality via silhouette. Using the recovered cluster structure from the first stage, we assign online trajectories following Algorithm 3.

---

**Algorithm 3** Stage 2: Anchored Assignment

---

**Require:** Online embeddings $Z_{\text{online}}$, recovered centroids $\{\mu_k^{\text{rec}}\}$, radii $\{r_k^{\text{rec}}\}$, threshold $\theta$
**Ensure:** Labels for online data, novel cluster IDs
 1: **for** each online embedding $z$ **do**
 2:     $d_k \leftarrow 1 - \frac{z^\top \mu_k^{\text{rec}}}{\|z\| \|\mu_k^{\text{rec}}\|}$ for all $k$
 3:     **if** $\min_k d_k \leq \theta \cdot r_k^{\text{rec}}$ **then**
 4:         Assign to nearest cluster: $\ell \leftarrow \arg\min_k d_k$
 5:     **else**
 6:         Mark as novel candidate
 7:     **end if**
 8: **end for**
 9: **// Sub-cluster novel candidates**
10: **if** n_novel_candidates $\geq m$ **then**
11:     Build $k$-NN graph on novel candidates
12:     Compute connected components
13:     **if** multiple components **then**
14:         Each component $\rightarrow$ separate novel cluster
15:     **else**
16:         Run Algorithm 1 with restricted grid search
17:     **end if**
18: **end if**
19: **return**  labels, novel_cluster_ids

---

The threshold $\theta$ controls sensitivity to novel modes. We set $\theta = 0.1$ on Reacher and Pusher, as clusters are well-separated, while we set $\theta = 0.05$ on Walker, as the continuous manifold creates the risk of over-assignment to existing clusters at mode boundaries. Nonetheless, the novelty threshold can be calculated in a data-driven manner based on the seen data, such as dynamically setting the novelty threshold as a high percentile (e.g., 99%) of the normalized distances of seen training data to their respective cluster centers. The recovered radius $r_k^{\text{rec}}$ is computed as the 95th percentile of within-cluster cosine distances, expanded by factor 1.2 to account for minor embedding shifts.

**Output Structure**    The two-stage procedure produces:

- **Recovered clusters** (IDs $0, \ldots, K_{\text{baseline}} - 1$): Correspond to baseline modes in the new space. Previously trained IRL agents are directly assigned.

- **Novel clusters** (IDs $\geq K_{\text{baseline}}$): Contain trajectories from genuinely new behavioral modes. New IRL agents are trained for these.

### D.3   Hyperparameters

| | Description | Value |
|---|---|---|
| $k_{\min}$ | Min neighbors for graph | $\max(5, N/50)$ |
| $k_{\max}$ | Max neighbors for graph | $\min(100, N/3)$ |
| $\sigma$ | RBF kernel bandwidth | 1.0 |
| $m$ | Min cluster size | $\max(5, 0.02 \cdot N)$ |
| $\alpha_{\text{behav}}$ | Jacobian weight | 0.3 |
| $\theta_{\text{novelty}}$ | Novelty threshold | 0.05–0.1 |
| $r_k^{95}$ exp | Factor for recovered radii | 1.2 |

Table 10: Graph clustering hyperparameters

# E   Additional Results

In this Appendix, we provide additional results for a more comprehensive overview.

## E.1   Nonparametric Bayesian IRL baseline

To evaluate how nonparametric Bayesian IRL approaches perform on high-dimensional continuous control tasks, we implement the transition-level BNP-IRL mixture framework Michini et al. (2015). Under a deep neural reward network formulation, a naive planning-in-the-loop setup requires running a Soft Value Iteration planner inside the inner Gibbs sampling loop to compute action likelihoods. With $N = 30,000$ transitions, this MDP solver loop requires $\approx 50$ hours per Gibbs iteration ($\approx 19.2$ days for 10 iterations), making it over $230\times$ slower than CoMI-IRL ($\approx 2$ hours total). To bypass this, we implement the standard planning-free formulation from the original continuous BNP-IRL framework (GPSRL) Michini et al. (2015), where subgoals are modeled as local feedback policies mapping states directly to actions to maximize demonstration likelihood.

Although this standard baseline runs efficiently, it fails on both reward learning and continuous control. NMI and ARI metrics collapse to 0.000 across all seeds in Reacher-v4, Pusher-v4, Hopper-v4, and HalfCheetah-v5, indicating a complete clustering collapse (Walker2d-v4 achieves only NMI = $0.069 \pm 0.053$, ARI = $0.038 \pm 0.064$). Furthermore, because the local feedback controllers lack value-based planning and long-term credit assignment, the deployed policies fail completely on locomotion tasks, obtaining negative or low ATR and failing to walk or maintain balance. Only exception is the Walker case, where the high standard deviation is given by the mode collapse into a staying still behavior.

We evaluate the baseline across 5 seeds (0–4) on five continuous control tasks. Table 11 reports NMI, ARI and ATR averaged over all seeds.

Table 11: Clustering Metrics for the Subgoal Baseline (Averaged across 5 seeds)

| Environment | Active Components ($K$) | NMI | ARI | ATR |
|---|---|---|---|---|
| Reacher-v4 | $1.0 \pm 0.0$ | $0.00 \pm 0.00$ | $0.00 \pm 0.00$ | $0.16 \pm 0.06$ |
| Pusher-v4 | $1.2 \pm 0.4$ | $0.00 \pm 0.00$ | $0.00 \pm 0.00$ | $-0.50 \pm 0.40$ |
| Walker2d-v4 | $3.8 \pm 0.7$ | $0.07 \pm 0.05$ | $0.04 \pm 0.06$ | $-0.86 \pm 1.87$ |
| Hopper-v4 | $1.0 \pm 0.0$ | $0.00 \pm 0.00$ | $0.00 \pm 0.00$ | $-0.18 \pm 0.22$ |
| HalfCheetah-v5 | $2.4 \pm 1.0$ | $0.00 \pm 0.00$ | $0.00 \pm 0.00$ | $-0.37 \pm 0.33$ |

## E.2   EPIC distance complete results

We hereby report the full EPIC distance analysis results, including all methods and all environments.

## E.3   Ablations

To validate the architectural choices of the pre-processing block composed by *RFF+MLP+CNN* (called preBlock in the results) within the BE, we run an ablation study over 3 seeds. To validate the loss composition of global contrastive, segment contrastive and DIM Hjelm et al. (2019) regularization, we run an ablations over 10 seeds, reporting in Table 14 the clustering quality of each version. While the versions with and without the DIM regularization are very similar (BE+CS and BE+DCS), we use the full version (BE+DCS) in our experiments as it showed higher stability and slightly better performance in most cases. Since Hopper and HalfCheetah are clearly separable, we skip their ablations for now.

Table 12: EPIC distance (Mean $\pm$ Std) to the true reward function. Comparisons and exact $p$-values are computed against the best-performing CoMI-IRL configuration (in bold) for each environment. Statistically non-significant differences ($p \geq 0.05$) are marked with [n.s.].

| Method | Pusher-v4 | Hopper-v4 | HalfCheetah-v5 | Walker2d-v4 |
|---|---|---|---|---|
| CoMI-IRL (AIRL) | $\mathbf{0.3360 \pm 0.1081}$ (Reference) | $\mathbf{0.5639 \pm 0.1691}$ (Reference) | $0.4409 \pm 0.0478$ ($p = 0.0059$) | $0.6134 \pm 0.0895$ ($p = 0.2676$ [n.s.]) |
| CoMI-IRL (GAIL) | $0.4938 \pm 0.0884$ ($p = 1.12 \times 10^{-7}$) | $0.7159 \pm 0.1445$ ($p = 0.0164$) | $\mathbf{0.3746 \pm 0.0673}$ (Reference) | $\mathbf{0.5759 \pm 0.0919}$ (Reference) |
| G-AIRL | $0.3470 \pm 0.1135$ ($p = 0.7061$ [n.s.]) | $0.6343 \pm 0.2171$ ($p = 0.3470$ [n.s.]) | $0.4356 \pm 0.0386$ ($p = 0.0074$) | $0.6356 \pm 0.0778$ ($p = 0.1040$ [n.s.]) |
| G-GAIL | $0.4963 \pm 0.0845$ ($p = 5.44 \times 10^{-8}$) | $0.7159 \pm 0.1445$ ($p = 0.0164$) | $0.3938 \pm 0.1013$ ($p = 0.5591$ [n.s.]) | $0.5944 \pm 0.0677$ ($p = 0.5783$ [n.s.]) |
| Ess-InfoGAIL (K=12) | $0.6528 \pm 0.0888$ ($p = 2.60 \times 10^{-18}$) | $0.7037 \pm 0.0848$ ($p = 0.0082$) | $0.7928 \pm 0.0950$ ($p = 1.20 \times 10^{-15}$) | $0.7338 \pm 0.0773$ ($p = 4.79 \times 10^{-6}$) |
| Ess-InfoGAIL (K=3) | $0.7603 \pm 0.1037$ ($p = 4.70 \times 10^{-13}$) | $0.7024 \pm 0.1737$ ($p = 0.0187$) | $0.6178 \pm 0.0854$ ($p = 7.37 \times 10^{-12}$) | $0.6785 \pm 0.0449$ ($p = 0.0009$) |
| Ess-InfoGAIL (K=6) | $0.6415 \pm 0.0702$ ($p = 2.60 \times 10^{-17}$) | $0.6848 \pm 0.1478$ ($p = 0.0234$) | $0.7011 \pm 0.1532$ ($p = 9.99 \times 10^{-17}$) | $0.6608 \pm 0.0575$ ($p = 0.0037$) |
| K-AIRL (K=12) | $0.3733 \pm 0.1160$ ($p = 0.1426$ [n.s.]) | $0.5846 \pm 0.1595$ ($p = 0.6811$ [n.s.]) | $0.4357 \pm 0.0739$ ($p = 0.0065$) | $0.5809 \pm 0.1064$ ($p = 0.8587$ [n.s.]) |
| K-AIRL (K=3) | $0.4285 \pm 0.1204$ ($p = 0.0221$) | $0.6343 \pm 0.2171$ ($p = 0.3470$ [n.s.]) | $0.4422 \pm 0.0589$ ($p = 0.0086$) | $0.6352 \pm 0.0712$ ($p = 0.0616$ [n.s.]) |
| K-AIRL (K=6) | $0.3470 \pm 0.1135$ ($p = 0.7061$ [n.s.]) | $0.5750 \pm 0.1613$ ($p = 0.8385$ [n.s.]) | $0.4447 \pm 0.0800$ ($p = 0.0051$) | $0.6104 \pm 0.1054$ ($p = 0.2705$ [n.s.]) |
| K-GAIL (K=12) | $0.4969 \pm 0.1190$ ($p = 2.74 \times 10^{-8}$) | $0.7185 \pm 0.1186$ ($p = 0.0047$) | $0.3585 \pm 0.1040$ ($p = 0.4809$ [n.s.]) | $0.5887 \pm 0.0996$ ($p = 0.6444$ [n.s.]) |
| K-GAIL (K=3) | $0.4341 \pm 0.1043$ ($p = 0.0079$) | $0.7159 \pm 0.1445$ ($p = 0.0164$) | $0.3872 \pm 0.1065$ ($p = 0.7102$ [n.s.]) | $0.5878 \pm 0.0581$ ($p = 0.6790$ [n.s.]) |
| K-GAIL (K=6) | $0.4963 \pm 0.0845$ ($p = 5.44 \times 10^{-8}$) | $0.7272 \pm 0.1257$ ($p = 0.0041$) | $0.4017 \pm 0.1053$ ($p = 0.3137$ [n.s.]) | $0.5993 \pm 0.0909$ ($p = 0.4297$ [n.s.]) |

### E.4    Jacobian Reweighting Effect

For each environment, we present the effects of applying the Jacobian reweighing of graph edges in terms of edge weight. While for Pusher and Reacher (Figure 9 and Figure 10) the impact is less prominent due to the already well-conditioned structure of the two spaces, so that the number of increased edges is very small (1% for Pusher and 5% for Reacher), in Walker (Figure 11) the reweighting has a higher effect, since it has more heterogeneous action effects on the different features. Between clusters, most edges are decreased, noticeable by the lighter color shade, while the 13% of increased edges are mostly within-cluster edges. On Hopper and HalfCheetah, where the average correlation is high and redundancy has been detected, we have no Jacobian reweighing of graph edges.

## F    Computational considerations

We ran our experiments on a 2020 MacBook Pro with 2 GHz Quad-Core Intel Core i5. In Section F, we report the average wall-clock times of Ess-InfoGAIL and CoMI-IRL (divided by phase) for each environment. It is important to notice that while the BE's training on GPU would take much less ($\sim$ 5 minutes), transformers training time also increases with the amount of training samples in the dataset.

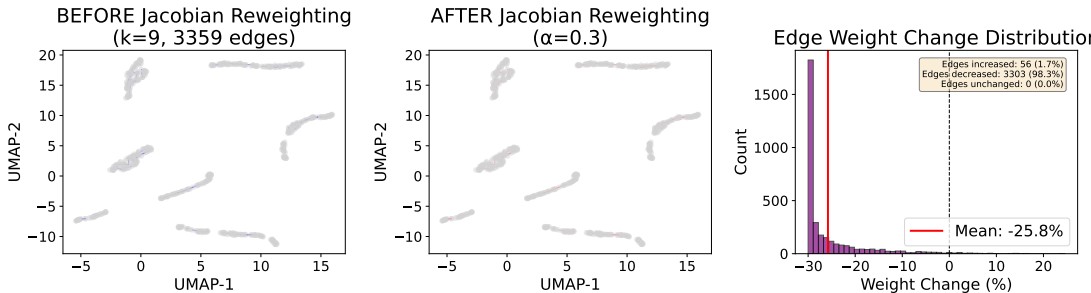

Figure 9: Jacobian Edge Reweighting on Pusher.

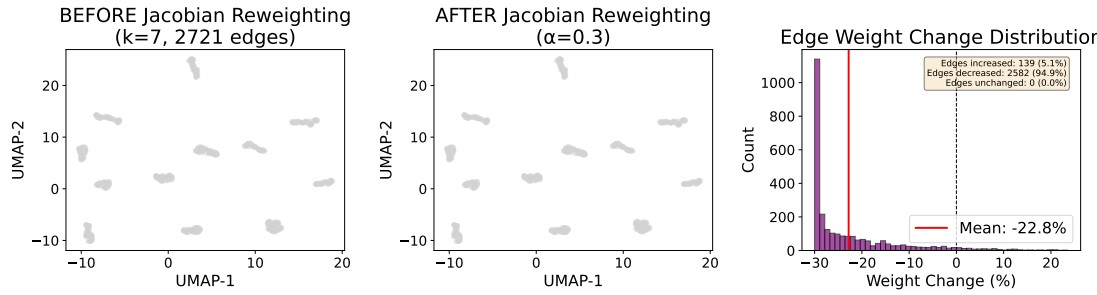

Figure 10: Jacobian Edge Reweighting on Reacher.

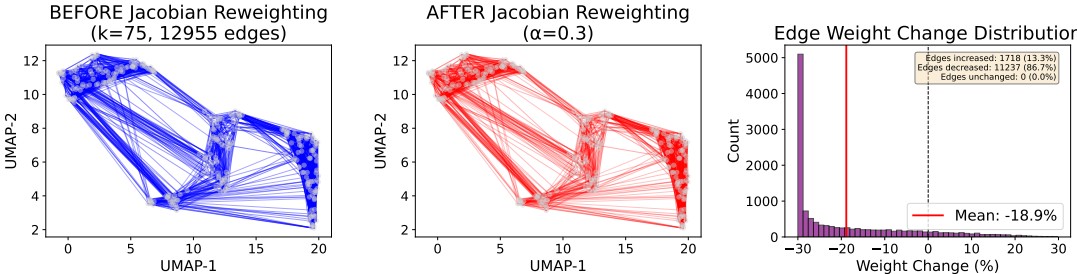

Figure 11: Jacobian Edge Reweighting on Walker.

Table 13: EPIC distance (Mean ± Std) for `Reacher-v4`. Single models are compared against CoMI-IRL (AIRL), and combined (max-reduced) models are compared against CoMI-IRL (AIRL) (Combined).

| Method | Type | EPIC Distance | $p$-value vs. Ref |
|---|---|---|---|
| **CoMI-IRL (AIRL)** | Single | **$0.7113 \pm 0.0274$** | (Reference) |
| **CoMI-IRL (AIRL) (Combined)** | Combined | **$0.5373 \pm 0.0595$** | (Reference) |
| CoMI-IRL (GAIL) | Single | $0.7762 \pm 0.0317$ | $8.03 \times 10^{-22}$ |
| CoMI-IRL (GAIL) (Combined) | Combined | $0.6165 \pm 0.0994$ | 0.0006 |
| Ess-InfoGAIL (K=12) | Single | $0.7060 \pm 0.1167$ | 0.7341 [n.s.] |
| Ess-InfoGAIL (K=12) (Combined) | Combined | $0.7109 \pm 0.1217$ | $2.00 \times 10^{-8}$ |
| Ess-InfoGAIL (K=3) | Single | $0.6598 \pm 0.0478$ | 0.0013 |
| Ess-InfoGAIL (K=6) | Single | $0.6207 \pm 0.0939$ | $1.59 \times 10^{-5}$ |
| G-AIRL | Single | $0.6858 \pm 0.0112$ | $2.88 \times 10^{-5}$ |
| G-GAIL | Single | $0.8302 \pm 0.0320$ | $3.92 \times 10^{-7}$ |
| K-AIRL (K=12) | Single | $0.6738 \pm 0.0635$ | $7.95 \times 10^{-5}$ |
| K-AIRL (K=12) (Combined) | Combined | $0.6305 \pm 0.0757$ | $7.10 \times 10^{-5}$ |
| K-AIRL (K=3) | Single | $0.6943 \pm 0.0251$ | 0.0356 |
| K-AIRL (K=6) | Single | $0.6759 \pm 0.0518$ | 0.0014 |
| K-AIRL (K=6) (Combined) | Combined | $0.6843 \pm 0.0196$ | $1.80 \times 10^{-8}$ |
| K-GAIL (K=12) | Single | $0.7342 \pm 0.0474$ | 0.0018 |
| K-GAIL (K=12) (Combined) | Combined | $0.6652 \pm 0.0561$ | $2.43 \times 10^{-9}$ |
| K-GAIL (K=3) | Single | $0.8121 \pm 0.0321$ | $1.27 \times 10^{-9}$ |
| K-GAIL (K=6) | Single | $0.7585 \pm 0.0267$ | $1.54 \times 10^{-10}$ |
| K-GAIL (K=6) (Combined) | Combined | $0.7105 \pm 0.0571$ | 0.0020 |

| | Reacher | Pusher | Walker | Hopper | HalfCheetah |
|---|---|---|---|---|---|
| **CoMI-IRL** | | | | | |
| BE | $\sim 16$ | $\sim 25$ | $\sim 17$ | $\sim 18$ | $\sim 18$ |
| Clustering (s) | $\sim 0$ | $\sim 0$ | $\sim 20$ | $\sim 0$ | $\sim 0$ |
| IRL | $\sim 14$ | $\sim 14$ | $\sim 48$ | $\sim 20$ | $\sim 48$ |
| Total | $\sim 30$ | $\sim 39$ | $\sim 65$ | $\sim 38$ | $\sim 66$ |
| **Ess-InfoGAIL** | | | | | |
| $K = 3$ | $\sim 35$ | $\sim 39$ | $\sim 55$ | $\sim 48$ | $\sim 38$ |
| $K = 6$ | $\sim 41$ | $\sim 46$ | $\sim 76$ | $\sim 57$ | $\sim 47$ |
| $K = 12$ | $\sim 48$ | $\sim 58$ | $\sim 103$ | $\sim 70$ | $\sim 58$ |

Table 15: Wall-clock time for the different components of CoMI-IRL and total, and the Ess-InfoGAIL experiments with different values of $K$. While all values are reported in minutes, the clustering values are reported in seconds. In CoMI-IRL's IRL phase, the reported time is the time of a single cluster, as it's fully parallelizable.

# G   Statistical Significance

In  Tables 16 to 20, we present the statistical significance pairwise comparison, ordered by significance according to the Wilcoxon signed-rank with Benjamini-Hochberg FDR correction, with $\alpha = 0.05$. We report the effect size through Cohen's $d_z$, where positive values indicate Method 1 outperforms Method 2. The comparisons between CoMI-IRL and Ess-InfoGAIL are bolded.

Table 14: **Ablation Study.** Impact of architecture (removing the *RFF+MLP+CNN* preprocessing) over 3 seeds and loss components on clustering quality over 10 seeds. Loss keys: **C**=Global Contrastive, **D**=DIM, **S**=Segment/Pairwise.

| Configuration | NMI | ARI | Sil |
|---|---|---|---|
| **Reacher-v4** | | | |
| *Architecture* | | | |
| BE (Full) | 0.84±0.00 | 0.63±0.00 | 0.87±0.06 |
| BE \ (preBlock) | 0.36±0.32 | 0.21±0.23 | 0.29±0.25 |
| *Loss Design* | | | |
| BE + C | 0.44±0.10 | 0.26±0.08 | 0.39±0.07 |
| BE + DC | 0.46±0.14 | 0.27±0.10 | 0.37±0.09 |
| BE + CS | 0.82±0.04 | 0.60±0.06 | 0.82±0.10 |
| BE + DCS | **0.83**±0.01 | **0.6**2±0.02 | **0.84**±0.06 |
| **Pusher-v4** | | | |
| *Architecture* | | | |
| BE (Full) | 1.0±0.00 | 1.0±0.00 | 0.99±0.01 |
| BE \ (preBlock) | 1.0±0.00 | 1.0±0.00 | 0.94±0.01 |
| *Loss Design* | | | |
| BE + C | 0.85±0.10 | 0.77±0.18 | 0.47±0.10 |
| BE + DC | 0.81±0.22 | 0.73±0.32 | 0.50±0.16 |
| BE + CS | 0.99±0.01 | 0.99±0.01 | 0.97±0.02 |
| BE + DCS | **1.0**±0.00 | **1.0**±0.00 | **0.98**±0.01 |
| **Walker2d-v4** | | | |
| *Architecture* | | | |
| BE (Full) | 0.75±0.05 | 0.77±0.07 | 0.60±0.01 |
| BE \ (preBlock) | 0.73±0.05 | 0.69±0.10 | 0.37±0.02 |
| *Loss Design* | | | |
| BE + C | 0.42±0.04 | 0.32±0.04 | 0.49±0.04 |
| BE + DC | 0.42±0.03 | 0.34±0.06 | 0.49±0.05 |
| BE + CS | **0.80**±0.06 | **0.82**±0.06 | 0.57±0.04 |
| BE + DCS | 0.79±0.06 | 0.80±0.10 | **0.57**±**0.03** |

| Method 1 | Method 2 | $p_{\text{FDR}}$ | $d_z$ |
|---|---|---|---|
| CoMI-IRL (AIRL) | G-AIRL | $2.5e-18$ | 13.02 |
| CoMI-IRL (AIRL) | K-GAIL(K=12) | $2.5e-18$ | 2.23 |
| CoMI-IRL (AIRL) | K-AIRL(K=6) | $2.5e-18$ | 3.93 |
| **CoMI-IRL (AIRL)** | **Ess-InfoGAIL(K=3)** | $2.5e-18$ | 4.52 |
| **CoMI-IRL (AIRL)** | **Ess-InfoGAIL(K=12)** | $2.5e-18$ | 2.83 |
| CoMI-IRL (AIRL) | K-GAIL(K=3) | $2.5e-18$ | 7.83 |
| CoMI-IRL (AIRL) | K-GAIL(K=6) | $2.5e-18$ | 4.03 |
| CoMI-IRL (AIRL) | G-GAIL | $2.5e-18$ | 11.93 |
| CoMI-IRL (AIRL) | K-AIRL(K=3) | $2.5e-18$ | 8.17 |
| CoMI-IRL (GAIL) | K-GAIL(K=12) | $2.5e-18$ | 2.24 |
| CoMI-IRL (GAIL) | K-AIRL(K=6) | $2.5e-18$ | 3.86 |
| CoMI-IRL (GAIL) | G-AIRL | $2.5e-18$ | 12.35 |
| **CoMI-IRL (GAIL)** | **Ess-InfoGAIL(K=3)** | $2.5e-18$ | 4.65 |
| **CoMI-IRL (GAIL)** | **Ess-InfoGAIL(K=12)** | $2.5e-18$ | 2.86 |
| CoMI-IRL (GAIL) | K-GAIL(K=3) | $2.5e-18$ | 8.20 |
| CoMI-IRL (GAIL) | K-GAIL(K=6) | $2.5e-18$ | 3.97 |
| CoMI-IRL (GAIL) | G-GAIL | $2.5e-18$ | 12.45 |
| CoMI-IRL (GAIL) | K-AIRL(K=3) | $2.5e-18$ | 8.61 |
| CoMI-IRL (AIRL) | K-AIRL(K=12) | $2.5e-18$ | 2.38 |
| CoMI-IRL (AIRL) | K-AIRL(K=12) | $2.6e-18$ | 2.37 |
| **CoMI-IRL (AIRL)** | **Ess-InfoGAIL(K=6)** | $2.9e-18$ | 2.43 |
| **CoMI-IRL (GAIL)** | **Ess-InfoGAIL(K=6)** | $4.4e-18$ | 2.37 |
| G-GAIL | K-AIRL(K=12) | $4.2e-12$ | −0.92 |
| K-GAIL(K=12) | G-GAIL | $5.5e-12$ | 0.95 |
| K-AIRL(K=12) | G-AIRL | $4.2e-11$ | 0.85 |
| K-GAIL(K=12) | G-AIRL | $6.0e-11$ | 0.87 |
| Ess-InfoGAIL(K=12) | K-GAIL(K=12) | $3.8e-9$ | −0.72 |
| Ess-InfoGAIL(K=12) | K-GAIL(K=12) | $5.8e-9$ | −0.71 |
| K-GAIL(K=12) | K-AIRL(K=3) | $1.2e-8$ | 0.70 |
| K-GAIL(K=3) | K-GAIL(K=12) | $1.4e-8$ | −0.70 |
| K-AIRL(K=3) | K-AIRL(K=12) | $3.7e-8$ | −0.67 |
| K-GAIL(K=3) | K-AIRL(K=12) | $5.1e-8$ | −0.67 |
| K-GAIL(K=6) | G-GAIL | $3.3e-7$ | 0.57 |
| G-GAIL | K-AIRL(K=6) | $3.4e-7$ | −0.56 |
| Ess-InfoGAIL(K=6) | G-GAIL | $5.3e-6$ | 0.49 |
| Ess-InfoGAIL(K=3) | K-GAIL(K=12) | $8.0e-6$ | −0.55 |
| Ess-InfoGAIL(K=6) | Ess-InfoGAIL(K=12) | $8.0e-6$ | 0.50 |
| Ess-InfoGAIL(K=3) | K-AIRL(K=12) | $9.4e-6$ | −0.52 |
| K-GAIL(K=6) | G-AIRL | $1.1e-5$ | 0.50 |
| K-AIRL(K=6) | G-AIRL | $1.4e-5$ | 0.49 |
| Ess-InfoGAIL(K=12) | K-GAIL(K=6) | $1.4e-5$ | −0.49 |
| Ess-InfoGAIL(K=12) | K-AIRL(K=6) | $1.6e-5$ | −0.49 |
| G-GAIL | K-AIRL(K=3) | $1.5e-4$ | −0.38 |
| Ess-InfoGAIL(K=6) | G-AIRL | $2.4e-4$ | 0.43 |
| Ess-InfoGAIL(K=3) | G-GAIL | $3.6e-4$ | 0.36 |
| K-GAIL(K=3) | G-GAIL | $5.1e-4$ | 0.36 |
| K-GAIL(K=6) | K-GAIL(K=12) | $7.6e-4$ | −0.36 |
| K-GAIL(K=12) | K-AIRL(K=6) | $8.9e-4$ | 0.36 |
| Ess-InfoGAIL(K=3) | Ess-InfoGAIL(K=12) | $1.1e-3$ | 0.34 |
| K-GAIL(K=6) | K-AIRL(K=3) | $1.2e-3$ | 0.33 |
| K-GAIL(K=3) | K-GAIL(K=6) | $1.6e-3$ | −0.33 |
| K-AIRL(K=3) | K-AIRL(K=6) | $1.7e-3$ | −0.32 |
| Ess-InfoGAIL(K=6) | K-GAIL(K=3) | $1.8e-3$ | 0.33 |
| Ess-InfoGAIL(K=6) | K-AIRL(K=3) | $2.0e-3$ | 0.33 |
| K-GAIL(K=3) | K-AIRL(K=6) | $2.4e-3$ | −0.32 |
| Ess-InfoGAIL(K=12) | K-AIRL(K=3) | $2.7e-3$ | −0.31 |
| Ess-InfoGAIL(K=12) | K-GAIL(K=3) | $2.7e-3$ | −0.31 |
| K-AIRL(K=3) | G-AIRL | $2.7e-3$ | 0.30 |
| K-AIRL(K=6) | K-AIRL(K=12) | $4.0e-3$ | -0.33 |
| K-GAIL(K=6) | K-AIRL(K=12) | $5.0e-3$ | -0.33 |
| K-GAIL(K=3) | G-AIRL | $6.9e-3$ | 0.28 |
| Ess-InfoGAIL(K=12) | G-AIRL | $1.3e-2$ | −0.23 |
| **Ess-InfoGAIL(K=3)** | **Ess-InfoGAIL(K=6)** | $1.7e-2$ | −0.24 |
| Ess-InfoGAIL(K=3) | G-AIRL | $2.0e-2$ | 0.26 |

Table 16: Pairwise agent comparison on Reacher-v4 (100 episodes/mode). Wilcoxon signed-rank test with Benjamini–Hochberg FDR correction ($\alpha = 0.05$). All rows shown have $p_{\text{FDR}} < 0.05$. $d_z$ is Cohen's $d_z$ (paired effect size); positive values indicate Method 1 outperforms Method 2.

| Method 1 | Method 2 | $p_{\text{FDR}}$ | $d_z$ |
|---|---|---|---|
| Ess-InfoGAIL(K=12) | G-AIRL | $5.7e-18$ | −2.67 |
| K-AIRL(K=3) | G-AIRL | $5.7e-18$ | −4.40 |
| **CoMI-IRL (AIRL)** | **Ess-InfoGAIL(K=6)** | $5.7e-18$ | 2.05 |
| CoMI-IRL (AIRL) | K-GAIL(K=3) | $5.7e-18$ | 3.85 |
| Ess-InfoGAIL(K=6) | G-AIRL | $5.7e-18$ | −2.07 |
| K-GAIL(K=3) | G-AIRL | $5.7e-18$ | −4.03 |
| **CoMI-IRL (AIRL)** | **Ess-InfoGAIL(K=12)** | $5.7e-18$ | 2.61 |
| **CoMI-IRL (AIRL)** | **Ess-InfoGAIL(K=3)** | $5.7e-18$ | 3.60 |
| Ess-InfoGAIL(K=3) | G-AIRL | $5.7e-18$ | −3.65 |
| CoMI-IRL (AIRL) | K-AIRL(K=3) | $6.4e-18$ | 4.19 |
| K-GAIL(K=6) | G-AIRL | $1.3e-17$ | −1.86 |
| CoMI-IRL (AIRL) | K-GAIL(K=6) | $1.8e-17$ | 1.83 |
| K-GAIL(K=12) | G-AIRL | $1.5e-16$ | −1.15 |
| CoMI-IRL (AIRL) | K-GAIL(K=12) | $4.2e-16$ | 1.15 |
| K-AIRL(K=6) | G-AIRL | $1.4e-15$ | −1.57 |
| CoMI-IRL (AIRL) | K-AIRL(K=6) | $8.9e-15$ | 1.55 |
| CoMI-IRL (AIRL) | G-AIRL | $5.3e-14$ | −0.72 |
| G-GAIL | G-AIRL | $6.7e-14$ | −0.68 |
| CoMI-IRL (AIRL) | CoMI-IRL (GAIL) | $4.0e-12$ | 0.70 |
| CoMI-IRL (AIRL) | G-GAIL | $4.6e-12$ | 0.67 |
| Ess-InfoGAIL(K=3) | K-AIRL(K=12) | $8.7e-12$ | −1.06 |
| K-GAIL(K=3) | K-AIRL(K=12) | $1.4e-11$ | −1.09 |
| Ess-InfoGAIL(K=12) | K-AIRL(K=12) | $5.5e-11$ | −0.90 |
| K-AIRL(K=12) | G-AIRL | $5.0e-10$ | −0.88 |
| K-GAIL(K=12) | K-AIRL(K=12) | $2.0e-8$ | −0.52 |
| K-AIRL(K=3) | K-AIRL(K=12) | $2.2e-8$ | −0.88 |
| K-GAIL(K=3) | G-GAIL | $2.3e-8$ | −0.63 |
| CoMI-IRL (GAIL) | K-GAIL(K=3) | $5.0e-8$ | 0.60 |
| CoMI-IRL (AIRL) | K-AIRL(K=12) | $5.2e-8$ | 0.86 |
| Ess-InfoGAIL(K=6) | K-AIRL(K=12) | $6.6e-8$ | −0.42 |
| Ess-InfoGAIL(K=3) | G-GAIL | $2.6e-7$ | −0.56 |
| **CoMI-IRL (GAIL)** | **Ess-InfoGAIL(K=3)** | $7.7e-7$ | 0.52 |
| K-GAIL(K=3) | K-AIRL(K=6) | $9.0e-7$ | −0.60 |
| Ess-InfoGAIL(K=3) | K-AIRL(K=6) | $1.0e-6$ | −0.58 |
| K-GAIL(K=6) | K-AIRL(K=12) | $1.6e-6$ | −0.55 |
| G-GAIL | K-AIRL(K=3) | $1.9e-6$ | 0.46 |
| K-GAIL(K=6) | G-GAIL | $3.2e-6$ | −0.44 |
| CoMI-IRL (GAIL) | K-AIRL(K=3) | $3.9e-6$ | 0.43 |
| Ess-InfoGAIL(K=6) | K-AIRL(K=6) | $5.2e-6$ | −0.43 |
| Ess-InfoGAIL(K=6) | K-AIRL(K=3) | $1.8e-5$ | 0.55 |
| **Ess-InfoGAIL(K=3)** | **Ess-InfoGAIL(K=6)** | $2.2e-5$ | −0.53 |
| CoMI-IRL (GAIL) | K-GAIL(K=6) | $2.8e-5$ | 0.40 |
| Ess-InfoGAIL(K=12) | G-GAIL | $4.0e-5$ | −0.43 |
| **CoMI-IRL (GAIL)** | **Ess-InfoGAIL(K=12)** | $7.1e-5$ | 0.40 |
| K-GAIL(K=12) | G-GAIL | $1.0e-4$ | −0.33 |
| K-GAIL(K=6) | K-AIRL(K=6) | $1.3e-4$ | −0.32 |
| CoMI-IRL (GAIL) | K-GAIL(K=12) | $2.8e-4$ | 0.30 |
| Ess-InfoGAIL(K=12) | K-AIRL(K=6) | $3.5e-4$ | −0.41 |
| K-AIRL(K=3) | K-AIRL(K=6) | $6.0e-4$ | −0.45 |
| Ess-InfoGAIL(K=6) | G-GAIL | $1.2e-3$ | −0.26 |
| **Ess-InfoGAIL(K=6)** | **Ess-InfoGAIL(K=12)** | $1.4e-3$ | 0.34 |
| K-GAIL(K=3) | K-GAIL(K=6) | $2.4e-3$ | −0.29 |
| **CoMI-IRL (GAIL)** | **Ess-InfoGAIL(K=6)** | $2.6e-3$ | 0.23 |
| K-GAIL(K=3) | K-GAIL(K=12) | $4.5e-3$ | −0.20 |

Table 17: Pairwise agent comparison on Pusher-v4 (100 episodes/mode). Wilcoxon signed-rank test with Benjamini–Hochberg FDR correction ($\alpha = 0.05$). All rows shown have $p_{\text{FDR}} < 0.05$. $d_z$ is Cohen's $d_z$ (paired effect size); positive values indicate Method 1 outperforms Method 2.

Table 19: Pairwise statistical comparisons for Hopper-v4 (Benjamini–Hochberg FDR correction, $\alpha = 0.05$). Only significant pairs shown ($p_{\text{FDR}} < 0.05$). $d_z$ = Cohen's $d_z$ (paired effect size).

| Method 1 | Method 2 | $p_{\text{FDR}}$ | $d_z$ |
|---|---|---|---|
| Ess-InfoGAIL(12) | G-GAIL | $9.28e{-}18$ | $-1.88$ |
| **CoMI-IRL (GAIL)** | **Ess-InfoGAIL(12)** | $9.28e{-}18$ | $1.88$ |
| Ess-InfoGAIL(3) | G-GAIL | $9.28e{-}18$ | $-2.77$ |
| **CoMI-IRL (GAIL)** | **Ess-InfoGAIL(3)** | $9.28e{-}18$ | $2.77$ |
| G-GAIL | K-AIRL(3) | $9.28e{-}18$ | $3.02$ |
| CoMI-IRL (GAIL) | K-AIRL(3) | $9.28e{-}18$ | $3.02$ |
| Ess-InfoGAIL(6) | G-GAIL | $9.81e{-}18$ | $-2.43$ |
| **CoMI-IRL (GAIL)** | **Ess-InfoGAIL(6)** | $9.81e{-}18$ | $2.43$ |
| CoMI-IRL (GAIL) | K-GAIL(3) | $3.81e{-}17$ | $2.42$ |
| K-GAIL(3) | G-GAIL | $5.21e{-}17$ | $-2.42$ |
| CoMI-IRL (GAIL) | K-AIRL(6) | $2.76e{-}16$ | $1.78$ |
| G-GAIL | K-AIRL(6) | $2.76e{-}16$ | $1.78$ |
| **CoMI-IRL (AIRL)** | **Ess-InfoGAIL(3)** | $6.46e{-}15$ | $1.42$ |
| K-GAIL(6) | G-GAIL | $6.06e{-}15$ | $-1.54$ |
| CoMI-IRL (GAIL) | K-GAIL(6) | $6.06e{-}15$ | $1.54$ |
| CoMI-IRL (GAIL) | G-AIRL | $6.06e{-}15$ | $0.71$ |
| G-GAIL | G-AIRL | $6.17e{-}15$ | $0.71$ |
| **CoMI-IRL (AIRL)** | **Ess-InfoGAIL(6)** | $2.05e{-}14$ | $1.29$ |
| K-AIRL(3) | G-GAIL | $6.79e{-}15$ | $-1.30$ |
| CoMI-IRL (AIRL) | K-GAIL(3) | $1.39e{-}14$ | $1.31$ |
| **CoMI-IRL (AIRL)** | **Ess-InfoGAIL(12)** | $1.24e{-}13$ | $1.07$ |
| Ess-InfoGAIL(3) | G-AIRL | $1.72e{-}13$ | $-1.10$ |
| K-GAIL(3) | G-AIRL | $6.39e{-}13$ | $-1.02$ |
| Ess-InfoGAIL(6) | G-AIRL | $2.00e{-}12$ | $-0.98$ |
| CoMI-IRL (AIRL) | K-AIRL(6) | $3.22e{-}12$ | $1.02$ |
| Ess-InfoGAIL(12) | G-AIRL | $3.24e{-}12$ | $-0.86$ |
| G-GAIL | K-AIRL(12) | $2.34e{-}11$ | $1.03$ |
| CoMI-IRL (GAIL) | K-AIRL(12) | $2.34e{-}11$ | $1.03$ |
| Ess-InfoGAIL(12) | K-GAIL(12) | $4.46e{-}11$ | $-0.73$ |
| CoMI-IRL (AIRL) | K-GAIL(6) | $5.40e{-}11$ | $0.91$ |
| K-AIRL(6) | G-AIRL | $2.31e{-}10$ | $-0.78$ |
| Ess-InfoGAIL(3) | K-GAIL(12) | $4.84e{-}10$ | $-0.63$ |
| CoMI-IRL (GAIL) | K-GAIL(12) | $2.23e{-}09$ | $0.89$ |
| K-GAIL(12) | G-GAIL | $2.30e{-}09$ | $-0.89$ |
| Ess-InfoGAIL(12) | K-AIRL(12) | $6.43e{-}09$ | $-0.57$ |
| K-GAIL(12) | K-AIRL(3) | $9.08e{-}09$ | $0.69$ |
| K-GAIL(6) | G-AIRL | $9.42e{-}09$ | $-0.67$ |
| Ess-InfoGAIL(3) | K-AIRL(12) | $1.04e{-}07$ | $-0.54$ |
| K-AIRL(3) | K-AIRL(12) | $1.40e{-}07$ | $-0.60$ |
| Ess-InfoGAIL(6) | K-GAIL(12) | $4.40e{-}07$ | $-0.51$ |
| CoMI-IRL (AIRL) | K-AIRL(12) | $5.45e{-}06$ | $0.53$ |
| Ess-InfoGAIL(6) | K-GAIL(6) | $1.40e{-}05$ | $-0.44$ |
| Ess-InfoGAIL(6) | K-AIRL(12) | $1.51e{-}05$ | $-0.44$ |
| Ess-InfoGAIL(3) | K-GAIL(6) | $2.85e{-}05$ | $-0.26$ |
| K-GAIL(3) | K-GAIL(12) | $2.85e{-}05$ | $-0.57$ |
| K-GAIL(3) | K-AIRL(12) | $2.86e{-}05$ | $-0.49$ |
| CoMI-IRL (AIRL) | K-GAIL(12) | $6.90e{-}05$ | $0.44$ |
| K-GAIL(6) | K-AIRL(3) | $4.34e{-}04$ | $0.31$ |
| K-AIRL(12) | G-AIRL | $1.22e{-}03$ | $-0.33$ |
| Ess-InfoGAIL(3) | K-GAIL(3) | $1.23e{-}03$ | $-0.29$ |
| Ess-InfoGAIL(6) | K-AIRL(6) | $2.75e{-}03$ | $-0.27$ |
| K-GAIL(3) | K-AIRL(3) | $3.80e{-}03$ | $0.22$ |
| Ess-InfoGAIL(3) | K-AIRL(6) | $3.80e{-}03$ | $-0.21$ |
| K-AIRL(3) | K-AIRL(6) | $5.18e{-}03$ | $-0.26$ |
| K-GAIL(12) | K-AIRL(6) | $5.42e{-}03$ | $0.38$ |
| K-AIRL(6) | K-AIRL(12) | $8.25e{-}03$ | $-0.31$ |
| CoMI-IRL (AIRL) | G-AIRL | $2.58e{-}02$ | $0.23$ |
| K-GAIL(6) | K-GAIL(12) | $3.09e{-}02$ | $-0.31$ |
| K-GAIL(6) | K-AIRL(12) | $3.48e{-}02$ | $-0.24$ |
| CoMI-IRL (GAIL) | G-GAIL | $3.82e{-}02$ | $-0.14$ |
| Ess-InfoGAIL(12) | K-AIRL(3) | $4.78e{-}02$ | $0.22$ |

Table 18: Pairwise statistical comparisons for Walker2d-v4 (Benjamini–Hochberg FDR correction, $\alpha = 0.05$). Only significant pairs shown ($p_{\text{FDR}} < 0.05$). $d_z$ = Cohen's $d_z$ (paired effect size).

| Method 1 | Method 2 | $p_{\text{FDR}}$ | $d_z$ |
|---|---|---|---|
| K-GAIL(6) | G-GAIL | $1.02e{-}11$ | $-0.34$ |
| Ess-InfoGAIL(12) | G-GAIL | $1.02e{-}11$ | $-0.43$ |
| Ess-InfoGAIL(3) | G-GAIL | $2.16e{-}10$ | $-0.80$ |
| K-GAIL(3) | G-GAIL | $2.62e{-}10$ | $-0.43$ |
| **CoMI-IRL (AIRL)** | **Ess-InfoGAIL(12)** | $2.87e{-}10$ | $0.47$ |
| Ess-InfoGAIL(6) | G-GAIL | $6.29e{-}10$ | $-0.33$ |
| **CoMI-IRL (AIRL)** | **Ess-InfoGAIL(3)** | $5.68e{-}9$ | $0.64$ |
| CoMI-IRL (AIRL) | K-GAIL(6) | $6.03e{-}9$ | $0.37$ |
| CoMI-IRL (AIRL) | K-AIRL(3) | $6.15e{-}9$ | $0.57$ |
| CoMI-IRL (AIRL) | K-GAIL(3) | $1.16e{-}8$ | $0.50$ |
| G-GAIL | K-AIRL(3) | $1.16e{-}8$ | $0.45$ |
| **CoMI-IRL (AIRL)** | **Ess-InfoGAIL(6)** | $2.57e{-}8$ | $0.39$ |
| G-GAIL | K-AIRL(12) | $5.96e{-}8$ | $0.41$ |
| K-GAIL(12) | G-GAIL | $1.55e{-}7$ | $-0.39$ |
| CoMI-IRL (AIRL) | K-AIRL(12) | $9.68e{-}7$ | $0.47$ |
| G-GAIL | K-AIRL(6) | $1.71e{-}6$ | $0.22$ |
| Ess-InfoGAIL(12) | G-AIRL | $3.53e{-}6$ | $-0.35$ |
| CoMI-IRL (AIRL) | K-AIRL(6) | $7.00e{-}6$ | $0.35$ |
| CoMI-IRL (AIRL) | G-AIRL | $8.64e{-}6$ | $0.53$ |
| G-GAIL | G-AIRL | $1.23e{-}5$ | $0.55$ |
| CoMI-IRL (AIRL) | K-GAIL(12) | $2.04e{-}5$ | $0.41$ |
| K-GAIL(6) | G-AIRL | $1.55e{-}4$ | $-0.30$ |
| Ess-InfoGAIL(3) | G-AIRL | $5.65e{-}4$ | $-0.31$ |
| Ess-InfoGAIL(6) | G-AIRL | $1.21e{-}3$ | $-0.24$ |
| K-GAIL(3) | G-AIRL | $1.97e{-}3$ | $-0.25$ |
| Ess-InfoGAIL(12) | K-AIRL(6) | $2.10e{-}3$ | $-0.21$ |
| K-AIRL(3) | G-AIRL | $2.22e{-}3$ | $-0.14$ |
| K-GAIL(12) | G-AIRL | $8.15e{-}3$ | $-0.35$ |
| CoMI-IRL (GAIL) | G-GAIL | $8.81e{-}3$ | $-0.40$ |
| **Ess-InfoGAIL(3)** | **Ess-InfoGAIL(12)** | $9.73e{-}3$ | $0.30$ |
| K-AIRL(12) | G-AIRL | $1.03e{-}2$ | $-0.31$ |
| CoMI-IRL (GAIL) | K-GAIL(6) | $1.42e{-}2$ | $0.01$ |
| Ess-InfoGAIL(12) | K-AIRL(3) | $1.50e{-}2$ | $-0.31$ |
| K-GAIL(6) | K-AIRL(6) | $2.00e{-}2$ | $-0.27$ |
| Ess-InfoGAIL(12) | K-GAIL(3) | $2.02e{-}2$ | $-0.22$ |
| **CoMI-IRL (AIRL)** | **CoMI-IRL (GAIL)** | $3.31e{-}2$ | $0.41$ |
| Ess-InfoGAIL(6) | K-AIRL(6) | $4.86e{-}2$ | $-0.12$ |

Table 20: Pairwise statistical comparisons for HalfCheetah-v4 (Benjamini–Hochberg FDR correction, $\alpha = 0.05$). Only significant pairs shown ($p_{\mathrm{FDR}} < 0.05$). $d_z$ = Cohen's $d_z$ (paired effect size).

| Method 1 | Method 2 | $p_{\mathrm{FDR}}$ | $d_z$ |
|---|---|---|---|
| **CoMI-IRL (GAIL)** | **Ess-InfoGAIL(12)** | $4.14e{-}18$ | $2.90$ |
| Ess-InfoGAIL(6) | G-GAIL | $4.14e{-}18$ | $-3.08$ |
| **CoMI-IRL (AIRL)** | **Ess-InfoGAIL(12)** | $4.14e{-}18$ | $3.59$ |
| **CoMI-IRL (AIRL)** | **Ess-InfoGAIL(6)** | $4.14e{-}18$ | $3.49$ |
| Ess-InfoGAIL(3) | G-GAIL | $4.14e{-}18$ | $-2.93$ |
| **CoMI-IRL (AIRL)** | **Ess-InfoGAIL(3)** | $4.14e{-}18$ | $3.35$ |
| **CoMI-IRL (GAIL)** | **Ess-InfoGAIL(3)** | $4.14e{-}18$ | $2.77$ |
| **CoMI-IRL (GAIL)** | **Ess-InfoGAIL(6)** | $4.14e{-}18$ | $2.83$ |
| Ess-InfoGAIL(6) | G-AIRL | $4.14e{-}18$ | $-3.06$ |
| Ess-InfoGAIL(12) | G-AIRL | $4.14e{-}18$ | $-3.28$ |
| Ess-InfoGAIL(12) | G-GAIL | $4.14e{-}18$ | $-3.30$ |
| Ess-InfoGAIL(3) | G-AIRL | $4.64e{-}18$ | $-2.78$ |
| K-GAIL(3) | G-GAIL | $2.01e{-}17$ | $-1.77$ |
| CoMI-IRL (AIRL) | K-GAIL(3) | $2.01e{-}17$ | $2.00$ |
| CoMI-IRL (AIRL) | K-AIRL(3) | $2.35e{-}17$ | $1.89$ |
| CoMI-IRL (GAIL) | K-GAIL(3) | $3.64e{-}17$ | $1.67$ |
| K-GAIL(3) | G-AIRL | $8.32e{-}17$ | $-1.69$ |
| K-AIRL(3) | G-AIRL | $9.80e{-}17$ | $-1.63$ |
| G-GAIL | K-AIRL(3) | $1.48e{-}16$ | $1.61$ |
| CoMI-IRL (AIRL) | K-AIRL(6) | $1.65e{-}16$ | $1.41$ |
| CoMI-IRL (AIRL) | K-GAIL(6) | $1.96e{-}16$ | $1.50$ |
| CoMI-IRL (GAIL) | K-AIRL(3) | $6.63e{-}16$ | $1.51$ |
| K-GAIL(6) | G-GAIL | $1.33e{-}15$ | $-1.31$ |
| CoMI-IRL (GAIL) | K-GAIL(6) | $3.04e{-}15$ | $1.24$ |
| K-GAIL(6) | G-AIRL | $7.60e{-}15$ | $-1.23$ |
| K-AIRL(6) | G-AIRL | $9.94e{-}15$ | $-1.17$ |
| G-GAIL | K-AIRL(6) | $7.00e{-}14$ | $1.16$ |
| CoMI-IRL (GAIL) | K-AIRL(6) | $1.16e{-}13$ | $1.11$ |
| Ess-InfoGAIL(6) | K-GAIL(12) | $2.74e{-}12$ | $-0.96$ |
| Ess-InfoGAIL(6) | K-AIRL(12) | $6.89e{-}12$ | $-0.97$ |
| Ess-InfoGAIL(3) | K-GAIL(12) | $9.59e{-}12$ | $-0.94$ |
| Ess-InfoGAIL(3) | K-AIRL(12) | $1.45e{-}11$ | $-0.94$ |
| CoMI-IRL (AIRL) | K-GAIL(12) | $1.84e{-}11$ | $0.93$ |
| CoMI-IRL (AIRL) | K-AIRL(12) | $2.49e{-}11$ | $0.86$ |
| Ess-InfoGAIL(12) | K-GAIL(12) | $3.53e{-}11$ | $-1.05$ |
| Ess-InfoGAIL(12) | K-AIRL(12) | $3.53e{-}11$ | $-1.05$ |
| CoMI-IRL (GAIL) | K-GAIL(12) | $2.55e{-}09$ | $0.73$ |
| K-GAIL(12) | G-AIRL | $4.41e{-}09$ | $-0.69$ |
| K-GAIL(12) | G-GAIL | $1.54e{-}08$ | $-0.77$ |
| G-GAIL | K-AIRL(12) | $1.36e{-}07$ | $0.67$ |
| K-AIRL(12) | G-AIRL | $1.65e{-}07$ | $-0.66$ |
| CoMI-IRL (GAIL) | K-AIRL(12) | $1.06e{-}06$ | $0.62$ |
| Ess-InfoGAIL(12) | K-AIRL(6) | $2.21e{-}06$ | $-0.59$ |
| Ess-InfoGAIL(3) | K-AIRL(6) | $2.21e{-}06$ | $-0.54$ |
| Ess-InfoGAIL(12) | K-GAIL(6) | $2.22e{-}06$ | $-0.56$ |
| CoMI-IRL (AIRL) | G-AIRL | $2.56e{-}06$ | $0.36$ |
| Ess-InfoGAIL(6) | K-GAIL(6) | $2.82e{-}06$ | $-0.59$ |
| Ess-InfoGAIL(6) | K-AIRL(6) | $2.82e{-}06$ | $-0.61$ |
| Ess-InfoGAIL(3) | K-GAIL(6) | $3.48e{-}06$ | $-0.52$ |
| K-GAIL(3) | K-AIRL(12) | $6.78e{-}06$ | $-0.55$ |
| K-GAIL(3) | K-GAIL(12) | $1.15e{-}05$ | $-0.53$ |
| K-AIRL(3) | K-AIRL(12) | $5.46e{-}05$ | $-0.52$ |
| K-GAIL(12) | K-AIRL(3) | $1.37e{-}04$ | $0.50$ |
| CoMI-IRL (AIRL) | G-GAIL | $2.74e{-}04$ | $0.38$ |
| **CoMI-IRL (AIRL)** | **CoMI-IRL (GAIL)** | $3.38e{-}04$ | $0.38$ |
| Ess-InfoGAIL(3) | K-GAIL(3) | $9.12e{-}04$ | $-0.38$ |
| Ess-InfoGAIL(3) | K-AIRL(3) | $1.10e{-}03$ | $-0.36$ |
| Ess-InfoGAIL(12) | K-GAIL(3) | $2.80e{-}03$ | $-0.37$ |
| Ess-InfoGAIL(12) | K-AIRL(3) | $3.83e{-}03$ | $-0.37$ |
| Ess-InfoGAIL(6) | K-GAIL(3) | $5.68e{-}03$ | $-0.34$ |
| K-GAIL(6) | K-AIRL(12) | $5.72e{-}03$ | $-0.32$ |
| Ess-InfoGAIL(6) | K-AIRL(3) | $7.20e{-}03$ | $-0.34$ |

