# OpenReview forum: "A Behavior-first approach to Multi-Intention Inverse Reinforcement Learning"
_TMLR — Under review for TMLR_

### Review · Reviewer_qQLY · 2026-07-03

**Summary Of Contributions:**

# Summary:

The paper addresses Multi-Intention IRL, where demonstrations come from several experts with different intentions. The authors frame prior work as reward-first (cluster by which reward likely generated a trajectory), which couples clustering with reward learning and fixes the number of modes $K$ a priori.

The authors propose CoMI-IRL, a behavior-first alternative that decouples the two. A Transformer encoder trained with contrastive losses produces trajectory embeddings, a graph-based method clusters them and infers $K$, and single-intention IRL (GAIL or AIRL) is run per cluster.

They also add adaptation to new behaviors via encoder finetuning with a stability regularizer and two-stage clustering. Evaluation covers five MuJoCo environments against Ess-InfoGAIL and clustering baselines, over five seeds, with Wilcoxon tests, FDR correction, and Cohen's $d_z$.


## Strengths:
- The reward-first versus behavior-first framing is clean and well motivated. Because clustering and reward learning are decoupled, the method can infer $K$ from data and adapt to new behaviors without full retraining, two things entangled methods cannot do.
- The controlled $K \neq K^\ast$ study (Table 1, Figures 2 to 6) is the strongest part of the paper, directly showing how fixed-$K$ methods degrade under both $K<K^\ast$ and $K>K^\ast$.
- The Reacher sub-mode result is a concrete and correctly-identified interpretability payoff, recovering twelve elbow-angle-consistent clusters where six were assumed, with low variance versus the entangled baseline (Tables 2 and 3).


## Weaknesses:

- The broad superiority claim rests on a single deep MI-IRL baseline (Ess-InfoGAIL). Alternatives named in the related work, including nonparametric Bayesian IRL that also infers $K$ are not compared, so the generality of "outperforms existing approaches" exceeds the evidence.
- The settings are toy-scale ($K^\ast \in \{3,6\}$), and several perfect scores (Pusher, Hopper, HalfCheetah in Table 1) land in environments the authors themselves call already separable, where naive K-Means with $K=K^\ast$ is also perfect. The distinct advantage is really shown only on Reacher and Walker2D.
- The "no $K$ needed" claim depends on unsupervised selection of many hyperparameters ($k$ for the kNN graph, Leiden resolution, novelty threshold $\theta$, loss coefficients $\alpha/\beta/\gamma/\delta$), but the selection procedure is under-specified, and it is not shown that it avoids using $K^\ast$ or labels.

**Audience:**

Yes

**Audience Explanation:**

The imitation-learning, IRL, and trajectory-representation parts of the TMLR audience would be interested. The reward-first versus behavior-first distinction is a useful lens, and the decoupled formulation infers the number of modes from data and adapts to new behaviors without full retraining, both of which entangled methods cannot do. The Reacher result, where a purely behavioral contrastive embedding surfaces finer-grained sub-modes than the reward-conditioned formulation, is worth knowing regardless of whether this specific architecture becomes standard.

**Broader Impact Concerns:**

I have no significant ethical concerns.

**Claims And Evidence:**

Yes

**Claims Explanation:**

The central claims are supported, with caveats I raise as requested changes rather than as grounds for rejection.

The main claim, that fixed-$K$ methods degrade when the assumed $K$ does not match the true $K^\ast$, is shown directly and convincingly. The controlled sweeps over $K<K^\ast$ and $K>K^\ast$ (Table 1, Figures 2 to 6) are the strongest part of the paper.
The second claim, that decoupling lets you inspect behavioral structure without reference to reward, is backed concretely by the Reacher analysis, where the recovered clusters track the elbow-joint configuration with low variance compared to the entangled baseline (Tables 2 and 3).
I also find the comparative gaps credible because they come with proper statistics (five seeds, Wilcoxon with FDR correction, effect sizes), which makes me reasonably confident they are not just seed noise.

Two places where the evidence does not yet reach the claim.
- First, the abstract and conclusion say CoMI-IRL "outperforms existing approaches," but Ess-InfoGAIL is the only deep baseline. Methods the authors themselves cite, including multi-modal GAIL, nonparametric Bayesian IRL that also infers $K$, and Zhu et al. 2024, are not compared, so a claim about the reward-first paradigm in general is broader than what one baseline can show.
- Second, several of the perfect scores (Pusher, Hopper, HalfCheetah in Table 1) occur exactly where plain K-Means with $K=K^\ast$ is also perfect, which means the method's distinct advantage is really demonstrated on Reacher and Walker2D. The narrative should say this plainly.

These are calibration and coverage issues, not misreported results.

**Requested Changes:**

- Broaden the deep baseline, or scope the claim. The superiority claim rests on Ess-InfoGAIL alone. It would be great if the comparison added at least one other method that also infers $K$ (a nonparametric Bayesian approach), or if the abstract and conclusion were scoped to the tested set. If continuous-space infeasibility is the reason for omitting these, please show it with a runtime or a failed run rather than assert it in one line.
- Specify how the hyperparameters and graph parameters are chosen without supervision, and confirm the choice does not use $K^\ast$ or labels. The "no $K$ needed" claim hinges on this. Could you describe exactly how $k$ (kNN), the Leiden resolution, the novelty threshold $\theta$, and the loss coefficients $\alpha/\beta/\gamma/\delta$ are set, confirm that ground-truth labels and $K^\ast$ play no role, and add a sensitivity analysis.
- Clarify the wallclock comparison (Appendix F). Please state whether the reported CoMI-IRL time includes contrastive encoder training and the k-resolution sweep on top of per-cluster IRL, so "comparable running time" is verifiable.
- Add at least one larger-$K^\ast$ setting (say $K^\ast \geq 10$). The current $K^\ast \in \{3,6\}$ is toy-scale, and the limitations section already flags scalability. I would view the mode-discovery and adaptation claims as much better supported with this.
- Fix the effect-size versus significance wording.

## Minor
- Sec 3.2: "in a an embedding space" -> "an".
- Sec 2 (EM-based Methods): "the performance highly depends on the how well these are selected" - "on how well".
- Sec 2 (Nonparametric Bayesian): "due to the computational limitations of apply Bayesian methods" - "of applying".
- Sec 5.2: "the number of increased edges is between 1%and5%" - missing spaces, "1% and 5%".
- Appendix A: "balacing in place" - "balancing".
- Appendix E.2: "most edges are decrease" - "decreased".
- Figures 9: the third-panel title is clipped at the right edge

---

> ### Author Response · Authors · 2026-07-15
>
> We thank the reviewer for their positive assessment and constructive suggestions. In this response, we would like to address the raised concerns and clarify any doubts.
>
> ***On outperforming existing approaches*** Regarding the existing nonparametric Bayesian IRL baselines, the application of BNIRL [1] in its continuous formulation (which avoids the planning-in-the-loop that makes the problem intractable, with an estimated single seed/environment runtime being 230x slower than a CoMI-IRL run) was shown to fail in locomotion tasks. This is likely because BNIRL looks for subgoals in a trajectory, returning very low NMI/ARI and ATR across all environments, also due to its closed-form local controller assumption, which makes the problem tractable but lacks the temporal reasoning required to coordinate continuous gaits in such tasks. We will add this to the Appendix.
>
>
> ***On the separability and CoMI-IRL advantages*** We present the environments which original spaces are separable to show that with a behavior-first perspective, even extremely simple methods like KMeans+GAIL/AIRL can perform comparably or outperform a reward-first method like EssInfoGAIL, given the correct knowledge of $K$. When the spaces are not easily separable or $K$ is unknown or misspecified and normal KMeans does not work, our proposed method reorganizes the space (as shown by high clustering results), leading to higher results. In the revised version, we will include a sentence explaining this more clearly.
>
> ***On not needing to know $K$ and the unsupervised selection procedure*** While there are indeed new load-bearing hyperparameters in the process, their range and selection (such as $k$ for the isolated components or the set of resolutions $\Gamma$) are fully automated and they are not fixed beforehand. While the novelty threshold in the adaptation process is fixed, it can be calculated in a data-driven manner based on the seen data, such as dynamically setting the novelty threshold as a high percentile (e.g., $99$%) of the normalized distances of seen training data to their respective cluster centers.
>
> ***On the wallclock comparison*** Yes, the "BE" in the table is the encoder training and the clustering includes the sweep (which applies specifically to Walker2d given its continuous manifold), which is completed in approximately 20 seconds.
>
> ***On larger scale experiments*** We thank the reviewer for suggesting this. We consider this a valuable future direction, currently out of the scope of the proposed work, as it deserves a more in-depth analysis and study of the scalability of the model in terms of mode discovery and in terms of the adaptation capabilities of the represented embedding space.
>
> Regarding the minor points and the effect-size wording, we thank the reviewer for pointing them out, and we will fix them in the revised version of the paper.
>
> [1] Michini, Bernard, et al. "Bayesian nonparametric reward learning from demonstration." IEEE Transactions on Robotics 31.2 (2015): 369-386.

---

### Review · Reviewer_dhmE · 2026-07-04

**Summary Of Contributions:**

The submission addresses the problem of Multi-Intention IRL, a setting in which we are given a dataset of trajectories which are generated by expert policies for different rewards, without knowledge of which reward/policy was followed in each trajectory.
The authors propose a method that first learns a representation of behavioral similarity based with a transformer-based model, then clusters trajectories based on this representation and finally uses off-the-shelf IRL methods for each cluster.
Experiments show that this performs better than baselines in some experiments.

Strengths:
 * Learning a behavior-similarity embedding first is an interesting approach and could be useful in some settings.

Weaknesses:
 * Arguably the paper is no longer addressing the Multi-Intention IRL setting, as it only focuses on identifying a highly performant policy, not on the quality of an extracted reward function. It is thus closer to imitation learning than a real IRL setting, making the whole framing of the work debatable.
 * The writing of the paper is overall not very clear. Many details are only available in the appendix or even entirely missing. For example, Section 4.1 never clearly mentions that the actual "encoder" is, except for being "transformer-based". Even from the appendix, it is unclear whether it is decoder-only, encoder-decoder or yet another structure. In another section "trajectory-level Jacobian features" are used, but how they are used is only stated in the appendix. It is also not clear how they are approximating "Jacobian" features beyond a rough heuristic.

**Audience:**

Yes

**Audience Explanation:**

Clustering first based on behavior and then using IRL methods could be helpful in some practical settings in which we only care about the quality of the extracted policy instead of the accuracy of the extracted reward

**Broader Impact Concerns:**

This work is purely algorithmic so I believe a broader impact statement is not strictly necessary.

**Claims And Evidence:**

No

**Claims Explanation:**

The paper does not seem to perform hyper-parameter tuning, neither of the proposed method nor of the baselines. As such the claim of outperforming baselines is not sufficiently supported. This is particularly problematic as the proposed method introduces numerous new hyper-parameters, for encoder structure, encoder loss, and clustering algorithm.

Arguably, the paper also does not address the IRL setting, as it considers the reward achieved by policies rather than the accuracy of the learned reward. It is thus closer to a behavior cloning / imitation learning setting than IRL.

Further, several implementation details are unclear both in the main text and in the appendix, in particular concerning the RFF/MLP/CNN structure of the encoder.

The number of random seeds evaluated per experiment is also relatively low for RL (5 each).

**Requested Changes:**

* The problem setting has to be clarified. What exactly is the goal? Do we care about accuracy of the extracted reward? Why or why not?
 * How were hyper-parameters chosen? How was the network structure including pre-processing chosen? If possible, hyperparameter-optimization should be used for baseline methods as well as the proposed method.
 * The main text should be more self-contained. Key details such as the Jacobian reweighting and its usage should be clearly defined.

These are all crucial to be clarified before this paper would be ready for publication in any venue.

Beyond that, there are some smaller issues/questions that could be clarified:
 * negative samples are not described in the text around (1)
 * Section 4.1 uses $S$ and $A$ ( state and action spaces) instead of $s$ and $a$ (states and actions)
 * Why is the CLS token prepended rather than appended, when it is meant to summarize the sequence?
 * The paper mentions a fallback to the Leiden-algorithm if the graph is fully connected. Does this happen in experiments, if yes, how often?

---

> ### Author Response · Authors · 2026-07-15
>
> We would like to thank the reviewer for highlighting the need of clarity for specific points, and we would like to address them in this response.
>
> ***On addressing MI-IRL*** We thank the reviewer for pointing this out (with a related point also raised by reviewer w5PN). While evaluating a learned reward by training and evaluating a policy is a very common practice (often called the ``rollout-method'' [1,2]), we acknowledge that a more direct analysis on the actual reward could help in making a stronger point and highlight our focus on allowing for specific reward recovery. Therefore, we used the EPIC pseudometric to analyze the relationship between the learned and recovered rewards, and it is reported in the following tables (that we will add to the paper in the revised version). The reported results showed significant p-values in pairwise comparison with the best-performing configuration in each environment. As EPIC in this case is calculated as a distance to the ground truth, lower is better. In this response we only include the main baselines for simplicity, but on the revised paper we include all the results:
>
> | Method              | HalfCheetah     | Hopper          | Walker2d        | Pusher          |
> |---------------------|-----------------|-----------------|-----------------|-----------------|
> | CoMI-IRL (AIRL)     | 0.44 $\pm$ 0.05 | 0.56 $\pm$ 0.17 | 0.61 $\pm$ 0.09 | 0.34 $\pm$ 0.11 |
> | CoMI-IRL (GAIL)     | 0.37 $\pm$ 0.07 | 0.72 $\pm$ 0.14 | 0.57 $\pm$ 0.09 | 0.49 $\pm$ 0.09 |
> | Ess-InfoGAIL (K=12) | 0.79 $\pm$ 0.09 | 0.70 $\pm$ 0.08 | 0.73 $\pm$ 0.08 | 0.65 $\pm$ 0.09 |
> | Ess-InfoGAIL (K=6)  | 0.70 $\pm$ 0.15 | 0.68 $\pm$ 0.15 | 0.66 $\pm$ 0.05 | 0.64 $\pm$ 0.07 |
> | Ess-InfoGAIL (K=3)  | 0.61 $\pm$ 0.08 | 0.70 $\pm$ 0.17 | 0.68 $\pm$ 0.05 | 0.76 $\pm$ 0.10 |
>
>
> On Reacher, as our method finds the submodes related to the elbow joint angle, we evaluate the distances between the original mode ground truth reward, the learned rewards for that mode separately and combined through union. In this case, we can see that when we consider the combination of the two learned rewards the distance with the ground truth is reduced for CoMI-IRL but not for Ess-InfoGAIL.
>
> | Method                        | EPIC            |
> |-------------------------------|-----------------|
> | CoMI-IRL (AIRL)               | 0.71 $\pm$ 0.03 |
> | CoMI-IRL (AIRL)(Combined)     | 0.54 $\pm$ 0.06 |
> | CoMI-IRL (GAIL)               | 0.77 $\pm$ 0.03 |
> | CoMI-IRL (GAIL)(Combined)     | 0.61 $\pm$ 0.10 |
> | Ess-InfoGAIL (K=12)           | 0.70 $\pm$ 0.11 |
> | Ess-InfoGAIL (K=12)(Combined) | 0.71 $\pm$ 0.12 |
> | Ess-InfoGAIL (K=6)            | 0.62 $\pm$ 0.10 |
> | Ess-InfoGAIL (K=3)            | 0.66 $\pm$ 0.05 |
>
> ***On hyperparameters tuning*** The tuning of most hyperparameters has been done manually. For Ess-InfoGAIL, we used the hyperparameters reported by the authors for the specific environments. We will clarify this with a sentence in the Appendix.
>
> ***On self containment*** We deferred more implementation-oriented information to the Appendix. Nonetheless, we will clarify the usage of the Jacobian reweighting in the main body of the revised version for clarity.
>
> ***On the encoder*** We would like to clarify that the encoder is indeed not an encoder-decoder or a decoder-only structure, but a transformer encoder-only structure. As our aim with this architecture is to construct an embedding space, the encoder-only is the best suited one.
>
> ***On the negative samples*** As we use dropout-augmented views, all the other trajectories in the batch are considered as negatives. Nonetheless, even if treated as negatives, their pushing away force is proportional to how different they are, so even if sometimes a ''positive'' trajectory is considered a ''negative'', it will not be pushed far away because its similarity is high (which explains why the clusters do not simply clump in one single point).
>
>
> ***On the $CLS$ token*** Prepending the $CLS$ token instead of appending it is also a common practice in the NLP community since the BERT architecture [3], as the first position allows for the accumulation of information on the trajectory as it absorbs information by attending all the other tokens.
>
> ***On the Leiden fallback*** The Leiden fallback happens only on continuous manifolds where isolated components are not automatically detected, such as in Walker2d.
>
> We will fix the typo in the revised version.
>
> [1] Gleave, Adam, et al. "Quantifying differences in reward functions." arXiv preprint arXiv:2006.13900 (2020).
>
> [2] Andrew Y. Ng and Stuart Russell. Algorithms for inverse reinforcement learning. In ICML, 2000.
>
> [3] Devlin, Jacob, et al. "Bert: Pre-training of deep bidirectional transformers for language understanding." Proceedings of the 2019 conference of the North American chapter of the association for computational linguistics: human language technologies, volume 1 (long and short papers). 2019.

---

### Review · Reviewer_w5PN · 2026-07-05

**Summary Of Contributions:**

# Summary

For the Multi-Intention IRL setting, this paper proposes a framework that first clusters trajectories according to behavioral similarity and then learns a separate reward function for each cluster. In other words, it performs behavior representation learning and clustering before applying IRL independently to each cluster, thereby decoupling clustering from reward learning. Experiments on multi-behavior MuJoCo tasks suggest that the method achieves favorable clustering results and downstream policy performance. The paper also provides evidence of interpretability and potential adaptability to new behaviors.

# Strengths

* The paper first discovers behavioral structure and then learns rewards, rather than defining trajectory similarity through reward likelihood. This provides a potentially useful new perspective for addressing Multi-Intention IRL problems.
* In traditional latent-conditioned MI-IRL methods, the latent code often serves both behavior clustering and reward/policy conditioning purposes. CoMI-IRL separates these two roles.
* The authors claim that behavioral modes can be automatically discovered through graph clustering, rather than requiring a pre-specified number of clusters as in Ess-InfoGAIL or the K-Means baseline.
* The trajectory embeddings learned from trajectory-level dynamic similarity can be visualized with UMAP, which offers a certain degree of interpretability.

# Weaknesses

* Although the authors claim that the method does not require specifying (K), it still appears to rely on several hyperparameters that play a role somewhat similar to (K), such as the graph clustering hyperparameters, resolution, and novelty threshold.
* The method first performs clustering to discover behavioral modes, but these modes may not necessarily correspond to the true intention or reward modes.
* In the Reacher task, six intentions are divided into twelve behavioral submodes, which further suggests that behavioral modes may not always align with the underlying intention or reward modes.
* The construction of positive and negative samples in the paper may deserve further discussion. The paper uses dropout-augmented views as positive pairs, while treating other trajectories in the batch as negatives. This means that different trajectories from the same behavior mode may still be treated as negatives in the loss.
* CoMI-IRL simplifies MI-IRL into single-intention IRL within each cluster. This formulation is clear, but it also introduces a potential cost: the more clusters there are, the more reward and policy models need to be trained. If a real-world scenario contains dozens or even hundreds of behavioral submodes, per-cluster IRL could become computationally expensive, and the limited data in small clusters may also affect reward learning.

**Audience:**

Yes

**Audience Explanation:**

Yes. The findings of this paper are likely to be of interest to researchers working on inverse reinforcement learning, representation learning for trajectories, unsupervised behavior discovery, and multi-modal decision-making.

The paper studies an important setting in IRL where demonstrations may come from multiple intentions or behavioral modes. Its main contribution is a behavior-first perspective: rather than using reward likelihood to cluster demonstrations, the proposed method first learns trajectory representations, clusters behaviors in the learned embedding space, and then performs reward learning separately within each discovered cluster. This is a conceptually clear and relevant alternative to reward-first MI-IRL methods.

**Broader Impact Concerns:**

The paper focuses on methodological advances in multi-intention inverse reinforcement learning and is evaluated on simulated continuous control benchmarks. I believe there's no specific ethical concerns about this work.

**Claims And Evidence:**

Yes

**Claims Explanation:**

I believe the answer should be Partially. \
The experiments provide convincing evidence that CoMI-IRL improves clustering quality and downstream policy performance on the evaluated MuJoCo benchmarks. The comparison with Ess-InfoGAIL and clustering-based baselines, together with ablations and statistical tests, supports the usefulness of the behavior-first decoupled pipeline. However, several broader claims are not fully supported. The method does not require a fixed latent K, but it still depends on graph-clustering hyperparameters, resolution selection, and novelty thresholds. The paper also conflates behavioral modes with intention/reward modes: clustering by trajectory dynamics may discover execution submodes rather than true intentions. In addition, ATR evaluates downstream policy performance but does not directly verify the recovered reward functions. The adaptation and interpretability claims are suggestive but lack stronger baselines and more systematic evidence. Thus, the claims are supported for the controlled benchmark setting, but they should be stated more narrowly.

**Requested Changes:**

# Proposed adjustments which are critical a recommendation for acceptance

- As discussed in the weaknesses, CoMI-IRL first clusters trajectories to identify behavioral modes. While this is a reasonable and interesting design choice, these behavioral modes may not always correspond exactly to the underlying intention or reward modes. It would be helpful if the authors could discuss this distinction more explicitly, and, if possible, provide additional empirical evidence showing when behavioral clusters align with intention/reward modes and when they may instead capture execution-level submodes.

- Since CoMI-IRL trains separate IRL agents for the discovered clusters, over-segmentation may increase computational cost and reduce the amount of data available for each per-cluster reward learner. This could affect both training stability and data efficiency when the number of discovered clusters becomes large. I encourage the authors to discuss this issue more systematically, especially in relation to scalability to many behavioral modes.

- The Jacobian reweighting component is potentially useful, but it also introduces additional handcrafted dynamics features and hyperparameters beyond the learned trajectory embeddings. It would strengthen the paper if the authors could provide a more systematic evaluation of this component, including its necessity, robustness, and sensitivity to hyperparameter choices.

- CoMI-IRL can be combined with either GAIL or AIRL, whereas Ess-InfoGAIL is based on GAIL and, as the authors note, is not straightforward to adapt to AIRL. This is understandable. However, when CoMI-IRL(AIRL) is directly compared with Ess-InfoGAIL, the observed performance gains may partly come from the AIRL backbone rather than solely from the proposed behavior-first paradigm. It would be useful for the authors to make the main comparisons under a consistent reward-learning backbone where possible, or to more clearly separate the effect of the backbone from the effect of the proposed decomposition.

- An interesting observation in the results is that, in environments where the original trajectory space is already relatively separable, such as Pusher, Hopper, and HalfCheetah, simple K-Means or graph-clustering baselines already achieve strong or even near-perfect NMI/ARI, and their ATR is close to that of CoMI-IRL. The paper also notes that decoupled baselines without learned embeddings perform well in clearly separable cases. It would be helpful if the authors could more clearly clarify when the proposed contrastive trajectory encoder is essential, as opposed to when the main benefit comes from the general behavior-first decoupling strategy.



# Proposed adjustments which would strengthen this work
- It would be valuable to evaluate the proposed method in settings with image observations and on newer benchmarks, such as OGBench. The current experiments mainly focus on low-dimensional MuJoCo-style continuous-control tasks. Additional experiments on visual inputs or more recent offline RL / goal-conditioned benchmarks would help demonstrate the generality of the proposed behavior-first representation and clustering framework.

- The paper states that CoMI-IRL can adapt to unseen behaviors without full retraining. However, the actual adaptation procedure still involves finetuning the encoder using both old and new data, adding a stability regularizer, re-clustering the seen data, and training new policies for novel clusters. Thus, the method is better understood as avoiding retraining the entire pipeline from scratch, rather than as a fully online, training-free, or memory-free incremental adaptation method. I suggest that the authors adjust the wording accordingly, and more clearly describe what parts of the system are reused, finetuned, re-clustered, or newly trained during adaptation.

---

> ### Author Response · Authors · 2026-07-15
>
> We would like to thank the reviewer for their positive assessment and for acknowledging the usefulness of the new perspective on approaching MI-IRL problems. In this response, we would like to answer and clarify the points raised as requested changes and weaknesses.
>
> ***On the difference between behavioral mode and intention***
> As we take a behavior-first approach, we focus on the behavioral aspects and patterns, thus the modes. For the experiments, this is reflected in how the model clusters the trajectories, leading to higher policy evaluations. Nonetheless, we performed additional analysis reporting the EPIC distances between the recovered rewards and the ground truth rewards (as reported later in the response). More specifically, the analysis over the Reacher environment shows a difference between the learned single-mode rewards and the ground truth reward. However, when considering the union of the learned single-mode rewards, the EPIC distance gets smaller. In this way, although the recovered reward might not in all cases represent the true intentions (assuming they are previously specified) they still reflect different behavioral patterns. Looking at a formulation such as in [1], this difference can be seen as a decomposition of the reward into a task component (the main pin the arm has to reach) and a style component (the angle of the elbow), indicating the relationship between the behavioral pattern and the reward, which can lead to different rewards encoding different behavioral modes.
>
> ***On oversegmentation*** Within well-separated spaces the segmentation is data-driven, as each isolated component is a coherent cluster. On environments with continuous manifolds, the oversegmentation problem may arise in the finetuning process (as seen with Walker2d). Nonetheless, and it will be focus of future studies, a subsequent analysis of the relationship between learned rewards is a viable option to potentially individuate oversegmentation cases (in a similar fashion to what has been done for Reacher). We will add a sentence in the limitations and future work to clarify this.
>
> ***On the Jacobian reweighting*** Our empirical analysis showed that reweighting only significantly impacts continuous manifold environments such as Walker. In the other environments, the impact is minimal (Reacher and Pusher) or the reweighting is not applied (HalfCheetah and Hopper) due to redundancy between embedding-based and Jacobian-based pairwise similarities.
>
> ***On the AIRL-GAIL differences and baselines performance*** In CoMI-IRL, AIRL and GAIL are applied to the exact same datasets, meaning the performance differences between CoMI-IRL versions come from the two paradigms. At the same time, by evaluating CoMI-IRL(GAIL) and Ess-InfoGAIL, their differences come from the new behavior-first perspective. Furthermore, the usefulness of the behavior-first perspective becomes clearer when comparing Ess-InfoGAIL with K-Means based methods on already separable spaces. With such perspective, in separable cases we often don't need more complex methods such as Ess-InfoGAIL as simple baselines can outperform it, showing exactly how such point of view can impact the learning.
> In the revised version, we will include a sentence explaining this more clearly.
>
> ***On the hyperparameters working as $K$*** While there are indeed new load-bearing hyperparameters in the process, their range and selection (such as $k$ for the isolated components or the set of resolutions $\Gamma$) are fully automated and they are not fixed beforehand. While the novelty threshold in the adaptation process is fixed, it can be calculated in a data-driven manner based on the seen data, such as dynamically setting the novelty threshold as a high percentile (e.g., $99$%) of the normalized distances of seen training data to their respective cluster centers.
>
> ***On the positive and negative samples***
> Indeed, different trajectories from the same behavior mode may still be treated as negatives in the loss. However, even if treated as negatives, they are pushed away proportionally to how different they are, so even if a ''positive'' trajectory is considered a ''negative'', it will not be pushed away too much because the similarity is high (which explains why the clusters do not simply clump in one single point).
>
> ***On the scalability costs***
> Scalability can become an issue in case of a very high number of clusters. Nonetheless, as every cluster is independent, through parallelization the execution times can be highly shortened. We will make this limitation clearer in the revised version of the paper.

---

> > ### Author Response · Authors · 2026-07-15
> >
> > ***On the verification of the recovered reward function***. We thank the reviewer for pointing this out (point also raised by reviewer dhmE). While evaluating a learned reward by training and evaluating a policy is a very common practice (often called the ``rollout-method'' [2,3]), we acknowledge that a more direct analysis of the actual reward could help in making a stronger point, and highlight our focus on allowing for specific reward recovery. Therefore, we used the EPIC pseudometric to analyze the relationship between the learned and recovered rewards, and it is reported in the following tables (that we will add to the paper in the revised version). The reported results showed significant p-values in pairwise comparison with the best-performing configuration in each environment. As EPIC in this case is calculated as a distance to the ground truth, lower is better. In this response, we only include the main baselines for simplicity, but in the revised version we will include all the results:
> >
> > | Method              | HalfCheetah     | Hopper          | Walker2d        | Pusher          |
> > |---------------------|-----------------|-----------------|-----------------|-----------------|
> > | CoMI-IRL (AIRL)     | 0.44 $\pm$ 0.05 | 0.56 $\pm$ 0.17 | 0.61 $\pm$ 0.09 | 0.34 $\pm$ 0.11 |
> > | CoMI-IRL (GAIL)     | 0.37 $\pm$ 0.07 | 0.72 $\pm$ 0.14 | 0.57 $\pm$ 0.09 | 0.49 $\pm$ 0.09 |
> > | Ess-InfoGAIL (K=12) | 0.79 $\pm$ 0.09 | 0.70 $\pm$ 0.08 | 0.73 $\pm$ 0.08 | 0.65 $\pm$ 0.09 |
> > | Ess-InfoGAIL (K=6)  | 0.70 $\pm$ 0.15 | 0.68 $\pm$ 0.15 | 0.66 $\pm$ 0.05 | 0.64 $\pm$ 0.07 |
> > | Ess-InfoGAIL (K=3)  | 0.61 $\pm$ 0.08 | 0.70 $\pm$ 0.17 | 0.68 $\pm$ 0.05 | 0.76 $\pm$ 0.10 |
> >
> > On Reacher, as our method finds the submodes related to the elbow joint angle, we evaluate the distances between the original mode ground truth reward, the learned rewards for that mode separately and combined through union. In this case, we can see that when we consider the combination of the two learned rewards the distance with the ground truth is reduced for CoMI-IRL but not for Ess-InfoGAIL.
> >
> > | Method                        | EPIC            |
> > |-------------------------------|-----------------|
> > | CoMI-IRL (AIRL)               | 0.71 $\pm$ 0.03 |
> > | CoMI-IRL (AIRL)(Combined)     | 0.54 $\pm$ 0.06 |
> > | CoMI-IRL (GAIL)               | 0.77 $\pm$ 0.03 |
> > | CoMI-IRL (GAIL)(Combined)     | 0.61 $\pm$ 0.10 |
> > | Ess-InfoGAIL (K=12)           | 0.70 $\pm$ 0.11 |
> > | Ess-InfoGAIL (K=12)(Combined) | 0.71 $\pm$ 0.12 |
> > | Ess-InfoGAIL (K=6)            | 0.62 $\pm$ 0.10 |
> > | Ess-InfoGAIL (K=3)            | 0.66 $\pm$ 0.05 |
> >
> > We will also narrow down the claims on the controlled benchmark, and adjust the wording for the adaptation method.
> >
> > [1] Chen, Letian, et al. "Fast lifelong adaptive inverse reinforcement learning from demonstrations." Conference on Robot Learning. PMLR, 2023.
> >
> > [2] Gleave, Adam, et al. "Quantifying differences in reward functions." arXiv preprint arXiv:2006.13900 (2020).
> >
> > [3] Andrew Y. Ng and Stuart Russell. Algorithms for inverse reinforcement learning. In ICML, 2000.

---

### Author Response · Authors · 2026-07-16

We would like to thank all the reviewers for your feedback and insights. We have tried our best to respond to your questions. We uploaded a revised version of our submission that incorporates your comments. With the modifications, the length of the paper went from 12 pages to 14. To facilitate comparison with the previous version, we highlighted the new parts in blue.